# Linking satellites to genes with machine learning to estimate phytoplankton community structure from space

Roy El Hourany[1], Juan Pierella Karlusich[2,3,4], Lucie Zinger[2,4,5], Hubert Loisel[1], Marina Levy[6], and Chris Bowler[2,4]

[1]Univ. Littoral Côte d'Opale, Univ. Lille, CNRS, IRD, UMR 8187, LOG, Laboratoire d'Océanologie et de Géosciences, F 62930 Wimereux, France
[2]Institut de Biologie de l'Ecole Normale Supérieure (IBENS), Ecole Normale Supérieure, CNRS, INSERM, Université PSL, 75005 Paris, France
[3]FAS Division of Science, Harvard University, Cambridge, MA
[4]Research Federation for the study of Global Ocean Systems Ecology and Evolution, FR2022/Tara GOSEE, 75016, Paris, France.
[5]Naturalis Biodiversity Center, 2300 RA Leiden, The Netherlands
[6]Sorbonne Université, LOCEAN-IPSL, Laboratoire d'Océanographie et du Climat ; Expérimentations et Approches Numériques, CNRS, IRD, MNHN, 75005 Paris, France

**Correspondence:** Roy El Hourany (roy.elhourany@univ-littoral.fr), Marina Levy (marina.levy@locean.ipsl.fr), Chris Bowler (cbowler@biologie.ens.fr)

**Abstract.** Ocean color remote sensing has been used for more than two decades to estimate primary productivity. Approaches have also been developed to disentangle phytoplankton community structure based on spectral data from space, in particular when combined with in situ measurements of photosynthetic pigments. Here, we propose a new ocean color algorithm to derive the relative cell abundance of seven phytoplankton groups, as well as their contribution to total chlorophyll-a (Chla) at the global scale. Our algorithm is based on machine learning and has been trained using remotely-sensed parameters (reflectance, backscattering, and attenuation coefficients at different wavelengths, plus temperature and Chla) combined with an omics-based biomarker developed using *Tara* Oceans data representing a single-copy gene encoding a component of the photosynthetic machinery that is present across all phytoplankton, including both prokaryotes and eukaryotes. It differs from previous methods which rely on diagnostic pigments to derive phytoplankton groups. Our methodology provides robust estimates of the phytoplankton community structure in terms of relative cell abundance and contribution to total Chla concentration. The newly generated datasets yield complementary information about different aspects of phytoplankton that are valuable for assessing the contributions of different phytoplankton groups to primary productivity and inferring community assembly processes. This makes remote sensing observations excellent tools to collect Essential Biodiversity Variables and provide a foundation for developing marine biodiversity forecasts.

## 1 Introduction

The production of organic matter (i.e., productivity) in marine ecosystems relies largely on phytoplankton. These unicellular photosynthetic microorganisms are evolutionarily diverse and exhibit a wide range of cell morphologies, sizes, photosynthetic

accessory pigments, elemental requirements, and biogeochemical and trophic functions (Pierella Karlusich et al., 2020). They play a key role in regulating ocean biogeochemistry (Fuhrman, 2009) and global climate, partly through the absorption of atmospheric CO2 and export of carbon to the deep ocean (Guidi et al., 2009; Tilman et al., 2014; Tara Ocean Foundation, 2022).

In order to investigate the potential impacts of environmental changes on marine ecosystem functioning (Ibarbalz et al., 2019; Henson et al., 2021), high-resolution, real-time, and global scale data on phytoplankton community structure are required (Pereira et al., 2013). However, existing knowledge about the global distribution of phytoplankton communities from in-situ observations is highly fragmented, spatially disparate, and temporally punctual. It is furthermore limited by both the challenges of in situ data collection and by the associated costs of measurement techniques, which range from microorganism imaging, and flow cytometry, to DNA sequencing (Hillebrand and Azovsky, 2001; Irigoien et al., 2004; Smith, 2007; Rodríguez-Ramos et al., 2015; Powell and Glazier, 2017; Righetti et al., 2019; Dutkiewicz et al., 2020; Pierella Karlusich et al., 2020).

Ocean color remote sensing offers an interesting alternative to map the global distribution of phytoplankton communities at the sea surface at a high spatio-temporal resolution. Since 1978, ocean color satellites have been used to observe the concentration of the main phytoplankton pigment, chlorophyll-a (Chla), considered as a proxy of phytoplankton biomass (O'Reilly et al., 1998; Sathyendranath et al., 2014). Recently, ocean color data have also been used to gain information about phytoplankton communities, such as their size structure, and their taxonomic or functional composition. This interest has facilitated the integration of the concept of phytoplankton functional types (PFT) and taxonomic groups (PG) into studies exploring various ecological and biogeochemical aspects (Le Quéré et al., 2005; Hood et al., 2006). Functional types refer to distinct categories associated with biogeochemical processes (e.g., silicifiers, calcifiers) and physiological adaptations to environmental factors (e.g., light, nutrients, turbulence), or to more practical categories identified through specific analytical techniques (e.g., pigment types) (IOCCG report N 14). On the other hand, phytoplankton groups correspond to taxonomic classes (e.g., diatoms, haptophytes, cyanobacteria). It is important to note that phytoplankton from different taxonomic groups can perform the same ecosystem function, e.g., both diatoms and silicoflagellates can biosilicify but represent different taxonomic groups. Specialized algorithms applied to ocean color data have consequently been developed to detect specific taxa with distinctive optical characteristics, e.g., Brown (1995) and Iglesias-Rodríguez et al. (2002), or the dominance of phytoplankton functional types e.g., Alvain et al. (2005), or the relative abundance of phytoplankton groups and size classes in term of their contribution to the Chla e.g., Hirata et al. (2011) and Xi et al. (2020, 2021) and lately, plankton assemblages and communities e.g., Kaneko et al. (2023), (Sathyendranath et al., 2014; Bracher et al., 2017; Mouw et al., 2017).

The diagnostic pigment analysis method (DPA, Vidussi et al. (2001)) relies on the association of secondary phytoplankton pigments with different broad taxonomic phytoplankton groups. DPA classification was later refined by Uitz et al. (2006) who gave different weightings to the diagnostic pigments to retrieve three phytoplankton size classes (PSC) from total Chla. The advantage of this method is that phytoplankton pigments can be measured in a cost-effective manner through high-performance liquid chromatography (HPLC). Today, large in-situ HPLC datasets are available with broad spatial and temporal coverage. These HPLC datasets have enabled the development of several DPA-based ocean color algorithms, which has made it possible to evaluate the abundance of different phytoplankton groups and size classes from ocean color satellite data e.g. Uitz et al.

(2006); Hirata et al. (2008, 2011); Soppa et al. (2014); Di Cicco et al. (2017); Organelli et al. (2013); El Hourany et al. (2019a, b); Brewin et al. (2010); Xi et al. (2021). However, the limitation of the DPA approach is that it is associated with large uncertainties in the classification of phytoplankton due to the presence of certain pigments in different phytoplankton taxa and cell size classes, which also vary with acclimation to light, temperature, and nutrient availability (Brewin et al., 2015; Chase et al., 2020).

In this work, we propose an alternate approach to develop an ocean color algorithm for phytoplankton group detection from in-situ metagenomic observations. The approach is ground-truthed on data collected by *Tara* Oceans, which constitutes the most comprehensive and harmonized molecular dataset available on phytoplankton taxonomic community structure on a global scale. More specifically, we used metagenomics reads to extract the global-scale distribution and abundance of the single-copy gene *psbO*, which is present across all phytoplankton groups and that provides an unbiased picture of phytoplankton cell abundances (Pierella Karlusich et al., 2022). We used these data, together with satellite-derived optical, physical, and biogeochemical parameters to train an unsupervised machine learning algorithm able to discern the non-linear relationship between phytoplankton taxonomic community structure and data derived from satellites. This new algorithm allowed us to derive the spatio-temporal variability of seven phytoplankton groups (PG) between 1997 and 2021. We then compared the performance of this new algorithm with that of two previous DPA-based algorithms (El Hourany et al., 2019a; Xi et al., 2021).

## 2 Materials

In this section, we present the datasets that were used for training the algorithm and for evaluating the outputs. The input dataset includes the in-situ distribution and abundance of phytoplankton groups inferred from metagenomics data from *Tara* Oceans and their associated satellite matchups. The outputs of the new algorithm are compared to a global dataset of in-situ HPLC diagnostic pigments, as well as with estimates from two DPA-based remote sensing algorithms.

### 2.1 Input dataset

#### 2.1.1 Metagenomic read abundance of the psbO gene

The *psbO* gene encodes the manganese-stabilizing protein, of around 270 amino acids, which constitutes a core subunit of photosystem II (PSII) and is unique to organisms carrying out oxygenic photosynthesis. The *psbO* gene is a single-copy gene in the vast majority of eukaryotes and prokaryotes. We used *psbO* reads from the metagenomes generated from the *Tara* Oceans expedition as a proxy of phytoplankton relative cell abundance (Pierella Karlusich et al., 2022). Among the 210 *Tara* Oceans stations, 145 stations sampled *psbO* reads in different ocean regimes from oligotrophic to eutrophic waters (Chla from 0.01 to 10 $mg.m^{-3}$, median at 0.3 $mg.m^{-3}$), from 2009 to 2013. Seawater samples were filtered in order to differentiate five planktonic size fractions (0.22-3 $\mu$m, 0.8-5 $\mu$m, 5-20 $\mu$m, 20-180 $\mu$m, 180-2000 $\mu$m). For the purpose of this study, we pooled the five size fractions into a single aggregated sample.

The *psbO* data enabled us to taxonomically differentiate seven phytoplankton groups: diatoms, dinoflagellates, green algae, haptophytes, pelagophytes, cryptophytes, and prokaryotes (Cyanobacteria) (Fig. 1). The *psbO* read abundances of these seven groups are expressed as relative phytoplankton cell abundance (%). Phytoplankton that were not assigned to any of these seven groups (Unclassified) represented less than 5% of the total relative cell abundance among all size classes.

The *psbO* measurements are proxies of relative cell abundance since this protein-encoding gene is generally present as a single-copy and is found in all phytoplankton groups. For example, if we take a huge diatom compared to a tiny *Synechococcus*, both have one *psbO* gene and therefore are counted as one within the *psbO* quantification. However, we know that a diatom's Chla content is way greater than that of Synechococcus (Agustí, 1991; Fujiki and Taguchi, 2002; Dairiki et al., 2020; Bock et al., 2022). This is where the conversion via size-dependent weights is essential in the case of Chla content estimation.

We should note however that filters may retain cells smaller than the nominal pore size because of net clogging, being trapped in fecal pellets, as well as being present as symbioses and colonies. This has been observed with prokaryotic pico-sized cells such as *Synechococcus* and Prochloroccocus being over-represented in the 180-2000 $\mu$m size fraction (Fig. 2). To minimize this impact, we based our size-weighting on 4 size-fractions, while excluding the 180-2000 $\mu$m size range. Chla fraction per group is expressed as follows:

$$\text{Chla fraction}_{\text{PG}} = \frac{\text{Chla}_{\text{in-situ}} \cdot \left( \sum_{s=1}^{4} (\text{psbO}_s^{\text{PG}} \cdot \text{size}_s) \right)}{\sum_{s=1}^{4} \sum_{\text{PG}=1}^{7} (\text{psbO}_s^{\text{PG}} \cdot \text{size}_s)} \tag{1}$$

where $psbO_s^{PG}$ is the *psbO* read abundance for a specific phytoplankton group (PG) and for one of the four size fractions (s), and size corresponds to the mid-value of the corresponding size range, following the protocol in Sommeria-Klein et al. (2021), i.e., x0.9 for the [0.6-1.2] size class, x2.9 for the [0.8-5] size class, x12.5 for the [5-20] size class, and x100 for the [20-180] size class. Applying equation 1 pools all size fractions per group while considering the *psbO* read values and the size factors mentioned above.

There are hence two levels of information derived from the molecular dataset; relative abundance of *psbO* reads as a proxy of relative cell abundance, and the fraction of Chla that each group represents. Both types of information have different implications. Chla is often used as a proxy of biomass, which is a relevant parameter for energy and matter fluxes (e.g., food webs, biogeochemical cycles), while cell abundance corresponds to species abundance for unicellular organisms, which is an important measure for inferring community assembly processes.

### 2.1.2 Satellite datasets

We used ocean color products from the GlobColour project (R2019, full archive reprocessed, 2020) from 1997 to the present day, downloaded from the GlobColour portal. These products were constructed by merging data from various satellite sensors: Sea-viewing Wide Field-of-view Sensor (SeaWiFS), Moderate Resolution Imaging Spectroradiometer (MODIS), Visible Infrared Imaging Radiometer Suite (VIIRS), Medium Resolution Imaging Spectrometer (MERIS), and Ocean and Land Colour Instrument (OLCI). We used sixteen GlobColour products: Chlorophyll-a concentration (Chla, product name: CHL1-AVW), Remote sensing reflectances (Rrs) at 11 wavelengths (412, 443, 469, 490, 510, 531, 547, 555, 620, 645, and 670 nm), light

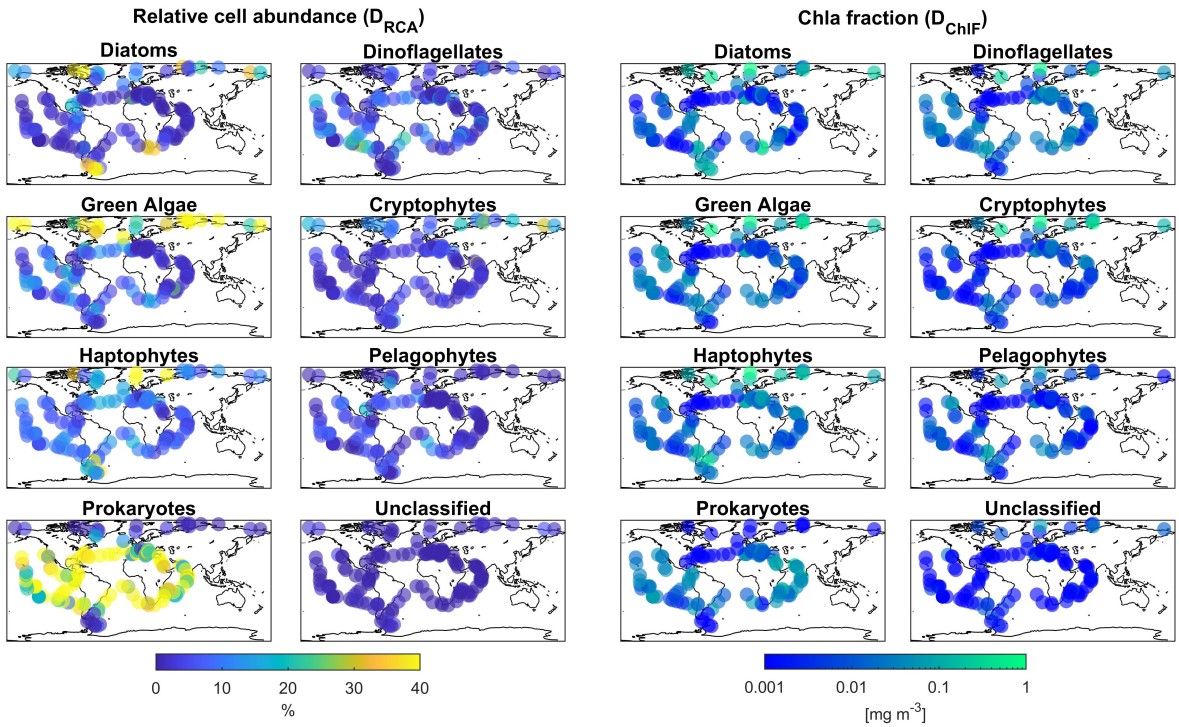

**Figure 1.** Global biogeographical patterns of marine phytoplankton relative cell abundance and Chla fraction per group based on *psbO* reads obtained from metagenomes from seawater samples collected during the *Tara* Oceans expeditions. Two sub-datasets are represented in this figure, the first, $D_{RCA}$ constituted of psbO-derived relative cell abundance and the second $D_{ChlF}$ *psbO*-derived Chla fraction per phytoplankton group.

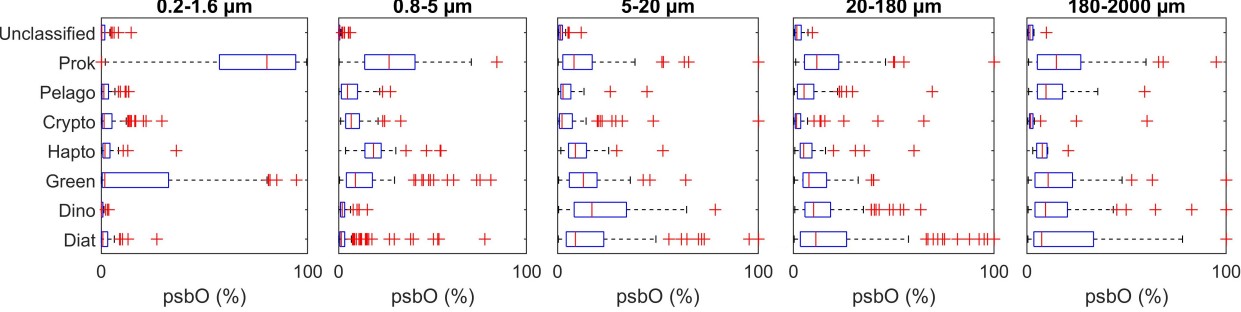

**Figure 2.** Relative abundance of *psbO* reads as a proxy of phytoplankton group cell abundance observed in each size fraction. The boxplots represent the distribution of each group and each panel shows the different size fractions. The equivalent plots for the psbO read values normalized by sequencing depth are displayed in supplementary Fig. S1'.

attenuation coefficient at 490 nm (Kd490), photosynthetically available radiation (PAR), Normalized fluorescence light height (NFLH) and particulate backscattering at 443 nm (bbp). These products have daily and 4km spatio-temporal resolution. In addition, we used the Climate Change Initiative (CCI) Sea Surface Temperature (SST) product at 4 km resolution and daily frequency distributed by the Copernicus Marine Services (CMEMS) portal.

## 2.2   HPLC datasets

To compare *psbO*-derived phytoplankton group distributions with more conventional, DPA-based products, we compiled a global HPLC dataset regrouping 12 000 HPLC observations from several HPLC datasets between 1997 and 2014 (Fig. 3): MAREDAT, NOMAD, SeaBASS, and other oceanographic campaigns: Labrador, Gep&co, Polarstern, BROKE-West, SAZ-Sense Voyage (Luo et al., 2012; Werdell and Bailey, 2005; Dandonneau et al., 2004; Bracher et al., 2015; Fragoso et al., 2016; Peloquin et al., 2013; Wright et al., 2010; de Salas et al., 2011). This HPLC dataset was collocated with satellite Glob-

Colour and the CCI SST product matchups. It depicts the abundance of the pigments most widely used to identify major phytoplankton groups: Fucoxanthin (Fuco), Peridinin (Perid), Alloxanthin (Allo), Zeaxanthin (Zea), Chlorophyll-b (Chlb), 19'-Hexanoyloxyfucoxanthin (19HF), and 19'-Butanoyloxyfucoxanthin (19BF) (Table 1). To estimate Chla fraction for each phytoplankton group, namely diatoms, dinoflagellates, haptophytes, green algae, cryptophytes, pelgophytes and prokaryotes, diagnostic pigments were used. The Chla fraction per group is expressed by:

$$\text{Chla}_{PG} = Chla_{in-situ} \cdot \frac{DP \cdot \alpha}{\sum DP \cdot \alpha} \tag{2}$$

where *a* is a coefficient associated with a diagnostic pigment (DP) for a specific PG.

Four sets of coefficients *a* are proposed for a global ocean application and are presented in Table 1 (Uitz et al., 2006; Soppa et al., 2014; Brewin et al., 2015; Losa et al., 2017). An examination of the values assigned to the coefficients by these four studies reveals disparities that do not consistently align across all pigments. Notably, while the coefficients for diatoms exhibit

similarity across the four sets, differences arise, for instance, in the case of dinoflagellates, only Uitz et al. (2006) and Brewin et al. (2015) show close coefficients associated to Perid, while in the case of haptophytes, where Brewin et al. (2015), Soppa et al. (2014), and Losa et al. (2017) estimates close coefficients attributed to 19HF. The discrepancies can be attributed to variations in the datasets utilized for coefficient estimation and differences in the methodologies employed. We chose to do an average of the output of the four sets of coefficients to increase the robustness of the results while considering the different

outputs of the utilization of these coefficients.

Simultaneously, *Tara* Oceans HPLC measurements (Pesant et al., 2015), which are available for the same stations and sampling time as for *psbO*, were considered to evaluate the correspondence between pigments and *psbO*-derived phytoplankton groups.

## 2.3   Phytoplankton groups satellite products

In order to compare the outputs of our method to those of existing DPA-based remote sensing algorithms, we used two previously published algorithms:

**Table 1.** Phytoplankton groups and size classes associated with their diagnostic pigments and coefficients $\alpha$.

| Phytoplankton size class | Phytoplankton group | Diagnostic Pigment (DP) | Uitz et al., 2006 | Soppa et al., 2014 | Brewin et al., 2015 | Losa et al., 2017 |
|---|---|---|---|---|---|---|
| **Micro** | **Diatoms**, Haptophytes, Chrysophytes, Di-noflagellates | Fucoxanthin (Fuco) (Jeffrey, 1980) | 1.41 | 1.55 | 1.51 | 1.27 |
| | **Dinoflagellates** | Peridinin (Perid) (Jeffrey, 1980; Jeffrey and Hallegraeff, 1987) | 1.41 | 0.41 | 1.35 | 2.43 |
| **Nano** | **Haptophytes**, Chryso-phytes, Dinoflagellates | 19'-Hexanoyloxyfucoxanthin (19HF) (Wright and Jeffrey, 1987) | 1.27 | 0.86 | 0.95 | 1.07 |
| | **Green algae**, Prasino-phytes | Chlorophyll-b (Chlb) (Vidussi et al., 2001) | 1.01 | 1.17 | 0.85 | 1.30 |
| | **Cryptophytes** | Alloxanthin (Allo) (Gieskes and Kraay, 1983) | 0.6 | 2.39 | 2.71 | 2.06 |
| | **Pelagophytes**, Hapto-phytes | 19'-Butanoyloxyfucoxanthin (19BF) (Wright and Jeffrey, 1987) | 0.35 | 1.06 | 1.27 | - |
| **Pico** | **Prokaryotes (Cyanobacteria)**, Green algae, Prasino-phytes, Chrysophytes, Euglenophytes | Zeaxanthin (Dandonneau et al., 2004; Guillard et al., 1985) | 0.86 | 2.04 | 0.93 | 2.36 |
| Coefficients based on global HPLC dataset corresponding to the sum of the weighted diagnostic pigments to the total Chla; Chla=$\sum \alpha DP$ | | | | | | |

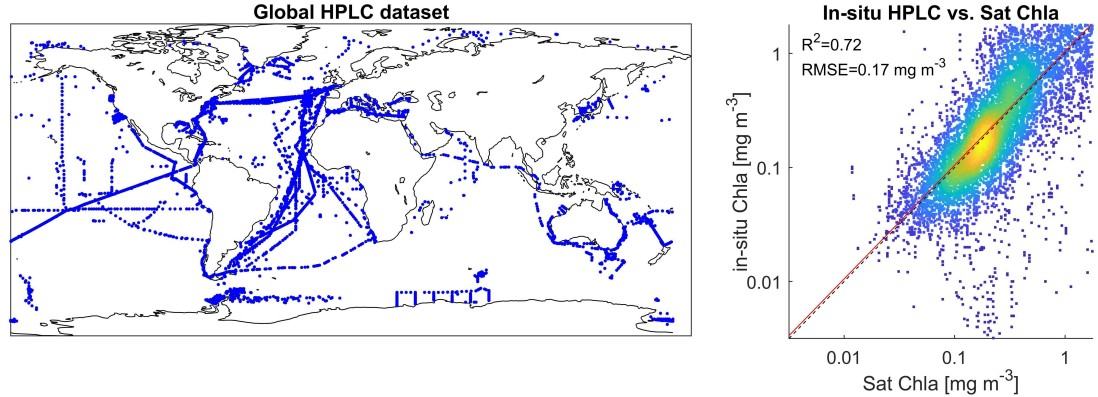

**Figure 3.** Geographical location of the global HPLC dataset stations regrouping observations from 1997 and 2014. The right panel represents a comparison between in-situ HPLC Chla measurement and its matchup using GlobColour Chla product.

### 2.3.1 CMEMS phytoplankton Chla fraction

This daily GlobColour product contains the concentration of each phytoplankton group (expressed in terms of Chla concentration fraction) based on the Xi et al. (2021) algorithm, from 2002 to the present at the global scale and with a 4km resolution. This algorithm estimates the Chla concentration of diatoms, dinoflagellates, haptophytes, green algae, and prokaryotes. The algorithm was implemented using HPLC-based phytoplankton groups using the DPA approach (Soppa et al., 2014) merged to ocean color (OC) Rrs products (412, 443, 490, 510, 531, 547, 555, 670, and 678 nm) and accounting for the influence of SST on the derived PG quantities. (product number: OCEANCOLOUR_GLO_BGC_L3_MY_009_103).

### 2.3.2 SOM phytoplankton pigments

SOM-Pigments (El Hourany et al., 2019a) is a machine learning-based algorithm that allows the estimation of phytoplankton pigment concentrations in oceanic waters from satellite ocean color data (Chla, Rrs at four wavelengths: 412, 443, 490, and 555nm) and SST. This algorithm is based on the use of Self-Organizing Maps (SOMs), an unsupervised neural network, and was calibrated using the HPLC dataset described above.

The SOM-Pigments algorithm applied to GlobColour products allows to estimate the daily concentration of ten phytoplankton pigments (Chlorophyll-a (Chla), Divinyl-Chlorophyll-a (DVChla), Chlorophyll-b (Chlb), Divinyl-Chlorophyll-b (DVChlb), 19'Hexfucoxanthin (19HF), 19'Butfucoxanthin (19BF), Fucoxanthin (Fuco), Peridinin (Perid), Alloxanthin (Allo), Zeaxanthin (Zea)) at the global scale from 1997 to the present and with a resolution of 4km. We then used the coefficients in Table 1 of Uitz et al. (2006),Soppa et al. (2014) and Brewin et al. (2015) to convert pigments into the Chla concentration of five phytoplankton groups, namely diatoms, dinoflagellates, haptophytes, green algae, and prokaryotes.

## 3 Methods

The algorithm we built to estimate phytoplankton groups from satellite data was built using SOM (Kohonen, 2013) and topology-constrained organization. This allowed us to confirm the non-linear relationships between phytoplankton group composition and satellite data through topology conservation. Next, we used the Ascending Hierarchical clustering algorithm to identify the large scale patterns generated by SOM. This allowed us to emphasize the predominant data structure learned by SOM and to characterize phytoplankton biomes. The steps of the training and operational phase of the SOM methodology were illustrated in flowcharts found in the supplementary materials (Fig. S2 to S4). Finally, to characterize the differences between the DPA- and *psbO*-based approaches, we used Random Forest models to highlight the cumulative importance of a pigment composition to estimate a phytoplankton group abundance. In the following section, each methodology and algorithm are explained in detail.

**Table 2.** Percentage of missing values within the initial database.

| *Tara Oceans* | *psbO* | | **Sat** | | | | | | |
|---|---|---|---|---|---|---|---|---|---|
| **D (145 stations)** | *Relative cell abundance* | Chla fraction per group | Chla | Rrs 412-709 nm | SST | bbp443 | Kd490 | NFLH | PAR |
| **Percentage of missing values** | - | 7% | 18% | 43-53% | 30% | 55% | 53% | 37% | 14% |

## 3.1 Structure of the training and test databases

The initial dataset (D) consists of the 145 *Tara* Oceans observations of *psbO* relative abundance of the seven defined phyto-plankton groups, the Chla fraction per group, and the associated matchups of 21 satellite-derived parameters (Chla, SST, Rrs at 15 wavelengths from 412 to 709nm, NFLH, Kd at 490m, PAR, and bbp at 443nm). The unclassified phytoplankton fraction was also considered, despite negligible values, to ensure coherence of the total phytoplankton pool. To extract the match-up for a given observation, a 3x3 pixel box was employed, centered around the observation's coordinates on the same day. The average of the non-outlier pixels was computed. If this approach was unproductive due to a low number of pixels within the 3x3 box or the absence of any pixel, a 3x3 pixel extraction was performed for the adjacent days (+1 and -1) (El Hourany et al., 2019a, b). Following these match-up exercises, we performed a baseline comparison between in-situ Chlorophyll-a (Chla) and satellite-derived Chla. This comparison is deemed satisfactory, with an average error rate of 33%.

We built two sub-datasets, the first ($D_{RCA}$) relating *psbO*-derived relative cell abundance of the seven defined phytoplankton groups to the 17 satellite-derived parameters, and the second ($D_{ChlF}$) joining *psbO*-derived Chla fraction per phytoplankton group and the same 17 satellite-derived parameters. We then constructed two algorithms, using either $D_{RCA}$ or $D_{ChlF}$, both based on the same SOM methodology described below. Following the positioning of *Tara* Ocean's stations, and the distribution of Chla values within both datasets (Fig. S5 and S6), both algorithms are suitable for case 1 water applications ( i.e. open ocean).

The rationale behind this is that the phytoplankton community should be treated as a whole; consequently, the variability of each phytoplankton group is dependent on each other in a relative way. $D_{RCA}$ and $D_{ChlF}$ both present missing values (Table 2), most likely due to cloud coverage or coastal/ice presence/proximity. In-situ *psbO*-based observations also contained missing values due to an absence of certain measurements at a given station. Since the in-situ dataset contains a low number of observations (145 stations), every observation is valuable. In order to overcome the several limitations faced with this training dataset, we used the SOM algorithm that can deal with missing values and allow a robust generalization in the case of limited observations (Jouini et al., 2013). Before applying the SOM, we ensured that all variables, phytoplankton observations, and satellite parameters were weighted alike while normalizing their values by their variance.

## 3.2 Self-Organizing map applied to *Tara* Oceans *psbO* data

### 3.2.1 General concept of SOM

The SOM algorithm is utilized for clustering multidimensional databases by assigning them to classes represented by a fixed network of neurons known as the SOM map. The SOM map consists of a rectangular grid of p x q neurons and defines a discrete

distance between neurons, enabling the partitioning of the dataset. Each cluster is associated with a neuron and represented by a prototype vector. Observations in the dataset are assigned to the nearest neuron based on the Euclidean Norm. A key feature of SOM is its ability to provide topological ordering, where close neurons on the map correspond to similar observations in the data space. The estimation of a neuron's vector and the topological order is determined through a minimization process of a cost function that depends on the distance between the neuron and its assigned observation. SOMs have been widely employed to complete missing data, utilizing the truncated distance (Folguera et al., 2015; Charantonis et al., 2015; Saitoh, 2016; Rejeb et al., 2022). The truncated distance is defined as a modification of the standard Euclidean distance between two observations that accounts only for the existing components of the vectors. This modification of the distance measure allows for the comparison of observations with incomplete information by considering only the existing components and effectively handling missing data. The SOM algorithm can then use this truncated distance measure in its learning process to complete missing data and integrate incomplete information, enabling more robust analysis and visualization of the data.

### 3.2.2   Training phase

Briefly, we first split the *Tara* Oceans *psbO* datasets so as to obtain 80% of the data to train the SOM, and 20% of the data as a test set, the latter consisting of 30 observations with complete *psbO* information. We did this separately for $D_{RCA}$ and $D_{ChlF}$ sub-datasets so as to generate SOMRCA, which stands for the algorithm specialized in relative cell abundance estimation, and SOMChlF for the algorithm specialized in Chla fraction per phytoplankton group.

During the SOM training, different combinations of satellite variables were used to determine the best set of variables to estimate the 7 phytoplankton groups in terms of relative cell abundance and Chla fraction. For each combination of variables, we increased the number of neurons from 10 to 1000 neurons, with an interval of 10 neurons, to determine the optimal size of the SOM. For each SOM obtained, we quantified quantization and topographic errors. The quantization error represents the difference between an observation and its closest neuron. This error is monitored during the training procedure until it reaches stability at a minimum value with increasing training epochs. This is where the training should stop to prevent overfitting. The quantization error is expressed as follows:

$$qe = \frac{1}{n} \sum_{i=1}^{n} \|x_i - w_{ci}\| \tag{3}$$

where $x_i$ is the vector of an input observation i; $w_{ci}$ is the vector of the closest neuron c of a sample $x_i$; n is the number of observations.

However, the topographic error is a representation of having, for each observation of the database, distant first and second best-matching neurons and is expressed as follows:

$$te = \frac{1}{n} \sum_{i=1}^{n} d(x_i) \tag{4}$$

where $d(x_i) = 1$ if the first and second closest neuron to $x_i$ are not adjacent, else $d(x_i)$=0.

Minimizing this quantity is important to ensure the preservation of the topological order within the SOM map with an increasing number of interpolated neurons. A one leave-out cross-validation procedure was performed to assign three performance metrics to help choose the best combination of SOM size and satellite variables: Regression coefficient ($R^2$) and Root-mean-squared error (RMSE). One should note that, for SOMChlF, the $R^2$ was calculated using log-transformed Chla values, and RMSE was calculated using real Chla values. At each iteration of the cross-validation procedure, we chose randomly one observation as a test, whereas the other observations served to train the SOM with the given grid size. We calculated the closest neuron to the test observation based on its satellite variables only and associated these latter with the neuron's seven phytoplankton groups vector. When all the observations were used as a test, we calculated a mean $R^2$ and an RMSE, associated with the given size map, while comparing the estimated and observed phytoplankton group values. The best SOM configuration and variable combination are based on an optimum where te, qe, and the RMSE are in low ranges while avoiding overfitting. The chosen SOM was tested using the 20% test set, providing independent performance metrics to evaluate the generalization of the chosen SOM. As a result, we present in the paper the performance metrics of the chosen SOM configuration based on the cross-validation procedure and the test set.

The optimal combination of satellite parameters for the SOMRCA and SOMChlF algorithms was determined to be Chla, SST, Rrs at four wavelengths (412, 443, 490, and 555 nm), bbp, and Kd490. The grid size for SOMRCA was set at 242 neurons, while SOMChlF had a grid size of 222 neurons. This selection was based on several factors, including a high regression coefficient between estimated and observed phytoplankton values, low error values of quantization and topographic error, and a low global RMSE encompassing all phytoplankton groups. The choice of Rrs bands aligns with previous work conducted on the PHYSAT method by Alvain et al. (2005) and Ben Mustapha et al. (2013). The PHYSAT method utilizes reflectance anomalies in the same four selected bands to identify dominant phytoplankton functional types. In the clear open ocean, the information contained in the remote sensing reflectance (Rrs) bands beyond 555 nm is limited due to the strong absorption by water (Torrecilla et al., 2011; Taylor et al., 2011). It should be noted that the Rrs bands selected are commonly measured by all sensors used to build the Rrs product of GlobColour. This overlapping of different sensors enhances data availability and coverage, thus increasing the importance of these Rrs bands within the initial dataset.

Through the iterative training process described above, the results show a significant increase in the general performance of the method when the number of neurons increases to a certain extent (Fig. 4). Using a number of neurons larger than the training dataset still allows a refined discretization. In this case, some neurons will capture a sample of the database, which permits to define a referent vector for these neurons. When the neuron did not capture any data observation, the discrete distance between the neighboring neurons was used to determine the referent vector w of each neuron that has not captured any data (Sarzeaud and Stephan, 2000; El Hourany et al., 2019a). This leads to preserving the topological order provided by new interpolated neurons. However, the quantization error's lowest values above 350 neurons might indicate overfitting.

### 3.2.3 Operational phase

During the operational phase, we estimated the phytoplankton group variability using the best combination of satellite parameters. The set of parameters of a pixel was projected onto the SOM. In doing so, the parameters at each pixel were normalized

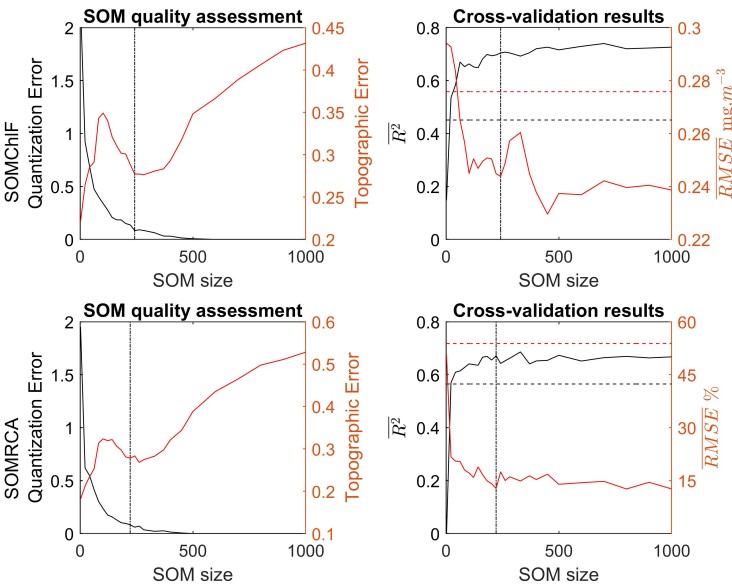

**Figure 4.** Quality assessment based on the quantization and topographic error related to the training of the SOMChlF and SOMRCA as a function of increasing SOM size (number of neurons) using Chla, SST, Rrs at 4 wavelengths (412, 443, 490 and 555 nm), bbp and Kd490. In parallel, the average regression coefficient and the root mean squared error as a function of increasing SOM size were calculated through a "one-leave out" cross-validation procedure. The dashed black and red lines correspond, respectively, to the $R^2$ and the RMSE using the "K-nearest neighbor" algorithm. Finally, the dotted lines correspond to the chosen SOM size for SOMChlF=242 neurons and SOMRCA=222 neurons.

by the variance of that same parameter within the initial training dataset to maintain an equal weight among the parameters and were assigned with the closest best-matching neuron using the truncated distance. At the end of the assignment phase, each pixel was associated with a referent vector corresponding to the best matching neuron, which includes the seven phytoplankton groups as a function of relative cell abundance in the case of SOMRCA, or Chla fraction in the case of SOMChlF. Since the training was undergone for the whole phytoplankton community at once, alongside the total Chla information, the SOM allows the inherent structure of the data to be preserved.

For this phase, level 3 mapped 4 km daily images were used to estimate the phytoplankton groups at the same spatio-temporal resolution.

### 3.2.4 Masking and uncertainty evaluation

Given that our initial dataset is of limited size, it is possible that it does not contain certain naturally occurring cases. In order to prevent abnormal predictions for cases not observed in the initial dataset, we conducted a quality evaluation of the method's output. This evaluation involved quantifying a reliability index by comparing the set of satellite parameters' values

at a particular pixel with the values of the same parameters in the initial dataset. If a satellite variable's value fell outside the range defined within the initial dataset by the mean value of the same variable's distribution plus or minus two standard deviations, it was considered distant. This evaluation was performed for all satellite variables per pixel, and the reliability index was determined by dividing the number of accepted variables by the total number of existing variables. A higher reliability index indicates greater reliability of the method, while regions with lower reliability index values require additional attention.

In the context of the global ocean, numerous uncertainties are associated with in-situ measurements, model parameterization, satellite parameters, and regions. The SOM algorithm is known to effectively reduce noise and mitigate the impact of uncertainties within the dataset (da Silva and Costa, 2013). However, the main source of uncertainty in the estimation process stems from selecting the best matching neuron. This involves finding and associating the closest neuron in the SOM with a new or unfamiliar observation, such as a satellite pixel. Due to the topology conservation, a pixel could be assigned to several close neurons, forming a neighborhood along a distance gradient. Consequently, a single satellite observation can represent various probabilities of phytoplankton group combinations influenced, to a certain extent, by the uncertainties of the satellite parameter.

To account for uncertainties in the estimations, we opted to associate each pixel and phytoplankton group (based on relative cell abundance or Chla fraction) with a weighted standard deviation derived from the values of the ten closest neurons. The weights were determined by the distances between the first ten matching neurons and the pixel. This approach allowed us to incorporate uncertainties into the assignment process and provide a confidence measure for each pixel's assignment. By considering both the reliability index and the weighted standard deviation, we could assess the influence of uncertainties in the satellite variables.

However, we should acknowledge the importance of addressing the uncertainties in the *psbO* measurements and their potential impacts on the algorithm's outputs, that are not taken into account in this study. This exclusion is primarily due to the absence of a comprehensive framework that accounts for all the associated steps in the quantification of *psbO*, including aspects such as filtration, extraction, and the accuracy of *psbO* analysis. Pierella Karlusich et al. (2022) conducted a thorough comparative study, evaluating *psbO* quantities against data obtained from confocal and optical microscopy, as well as cytometry, revealing an agreement of 70% (Spearman's Rho =0.64–0.71, p-value <.001). However, it is essential to recognize that like *psbO*, every quantification method is subject to uncertainties stemming from the various steps of the quantification process, emphasizing the necessity of comprehensive assessments within every in-situ measurement protocol.

### 3.3 Characterisation of phytoplankton biomes

To emphasize the predominant data structure learned by SOMChlF, the Ascending Hierarchical Clustering algorithm (AHC) was used to characterize phytoplankton biomes on the basis of their Chla fractions (a proxy of a phytoplankton group's biomass) and optical signature.

The HAC is a bottom-up clustering algorithm. The HAC starts with individuals and combines them according to their similarity (with respect to the chosen distance) to obtain new clusters. The exact number of biomes is not known a priori but at the end of the SOM+HAC procedure, several possibilities of a number of clusters to be taken into account were revealed.

A compromise was made between the number of clusters we could explain from a physical point of view and the number of clusters for which we needed to include the maximum of information embedded in the dataset. This procedure has been used with success in several studies (Reygondeau et al., 2014; Richardson et al., 2003; Rossi et al., 2014; Sawadogo et al., 2009; El Hourany et al., 2021). At the end of the HAC clustering phase, each neuron of the SOMChlF was associated with a cluster. The association of several neurons in a cluster allows us to identify common phytoplankton community structures, and therefore characterize phytoplankton biomes. Upon applying SOMChlF as described in the operational phase section, each pixel of a satellite image could be associated with a cluster.

## 3.4 Evaluation of pigments to estimate phytoplankton groups

Each phytoplankton group's *psbO* abundance was associated with its corresponding HPLC pigment measurements performed on the same *Tara* Oceans station. The ability of pigments to predict a specific phytoplankton group was evaluated using a bagged random forest algorithm (number of learners set to 200), following the permutation-based importance method.

Using this method, a pigment composition of the seven major phytoplankton pigments cited in Table 1 was tested to predict the abundance of each of the seven *psbO*-derived phytoplankton groups, and therefore estimate their importance relative to each group. The concentration of each pigment was converted in terms of pigment ratios, a ratio relative to the sum of all pigment concentrations, and in parallel, the *psbO*-derived relative abundance was used.

The bagged random forest algorithm is a set of decision trees, each constituted of internal nodes and leaves. Within the internal nodes, the algorithm uses pigment data as the predictor variable to partition the dataset into subsets based on pigment characteristics. These subsets are then utilized to predict the abundance of specific phytoplankton groups, enabling effective analysis of the importance of pigments to describe the variability of a phytoplankton group. Since this algorithm is used in a case of regression, the training is done while minimizing the error between the *psbO*-derived phytoplankton group abundance and the predicted one. The permutation-based importance method will randomly shuffle each pigment and compute the change in the model's performance to predict the abundance of a phytoplankton group.

## 4 Results and discussion

### 4.1 Performances, uncertainties and spatial limitation of the SOMRCA and SOMChlF algorithms

To assess the integrity of inter-variable relationships within the input data represented by the Self-Organizing Map (SOM), a comparison of correlation coefficients and distributions of phytoplankton group values was conducted between SOMRCA and SOMChlF with their respective measures, $D_{RCA}$ and $D_{ChlF}$. This analysis indicated that the correlation coefficients and value distributions remained unaffected within both SOMRCA and SOMChlF compared to the initial dataset, illustrating the capacity of SOM to retain the characteristics of the original dataset post-training (Fig. S5 and S6).

The cross-validation and test exercises demonstrated an average $R^2$ of 0.68 for SOMRCA and 0.74 for SOMChlF across all phytoplankton groups (Fig. 5, table 3). Aggregating all Chla fractions showcased a satisfactory agreement between estimated

total Chla and in-situ values ($R^2$= 0.83), indicating the preservation of the initial phytoplankton quantity expressed in total Chla. For SOMRCA, the RMSE ranged between 2% and 24% in the test set and between 2% and 19% in cross-validation. The highest errors were observed for Prokaryotes, reaching 24% due to their high relative cell abundance in the initial dataset. In the case of SOMChlF, the RMSE ranged between 0.02 and 0.24 $mg.m^{-3}$ in cross-validation and 0.02 and 0.31 in the test set, with the highest error associated with the estimation of Chla, stemming from the cumulative Chla fractions of phytoplankton groups. Notably, the largest RMSE among phytoplankton groups was observed for Diatoms' Chla fraction, attributed to their substantial Chla content and its exponential relationship with total Chla. The MRD highlighted a distinct contrast between SOMRCA and SOMChlF performance. Notably, SOMRCA exhibited a significantly higher median relative deviation, approximately three times that of SOMChlF's MRD. The MRD for SOMRCA fluctuated between 0.36 and 0.81 for cross-validation and between 0.28 and 0.92 for the test set, with Dinoflagellates exhibiting the highest MRD. In contrast, SOMChlF's MRD per group ranged between 0.13 and 0.24 for phytoplankton Chla fraction and 0.33 for Chla in the test set. This discrepancy emphasizes the complexity of determining the phytoplankton community structure in terms of relative cell abundance, indicating the likelihood of diverse community structures responding to the same satellite-derived environmental context.

Given the limited size of the initial dataset, applying SOMRCA and SOMChlF to the global satellite data must be done with caution. For each pixel and at each time step between 1997 to 2021, we performed the quality control described in section 3.2.3 to provide a measure of the applicability of this method (Fig. 6). Regions of low confidence can be identified where the value of the reliability index does not exceed 60% throughout the time series (Threshold arbitrarily chosen while evaluating the frequency histogram of this index's values in Fig. 6. A value of 60% roughly translates to the exclusion of 3 out of 8 satellite parameters' values considered outliers at a certain pixel). These regions are mainly found in coastal and turbid waters, as well as the South Pacific Ocean gyre, and are characterized either by very high or very low Chla values. This result is expected because the SOM algorithm is mainly adapted for case 1 waters and cannot extrapolate beyond the distribution of values in the initial dataset. Furthermore, moderate confidence regions in which around 20% of the pixels fall out of the accepted bounds, are highlighted by a reliability index under 80%. These regions are mainly found at high latitudes, especially in the Southern Ocean, mainly due to the limited number of available samples in the area and the particular optical characteristics of that region (Mitchell et al., 1991).

Uncertainty values reached 30% relative cell abundance for SOMRCA and 0.15 $mg.m^{-3}$ of Chla for SOMChlF, revealing distinct regional patterns in both cases. Notably, the observed uncertainties generally aligned with the concentration gradient in Chla fraction and cell abundance per group. The uncertainty associated with SOMRCA's outputs corresponded to the high relative deviation noted in the test and cross-validation, suggesting the potential acceptance of multiple community structures represented by the neurons of SOMRCA for a single satellite pixel, thus contributing to increased uncertainty levels. Regions at high latitudes exhibited the highest uncertainties for diatoms, green algae, and haptophyte relative cell abundances, while the Southern Ocean displayed heightened uncertainties specifically for prokaryotic cell abundance.

The increased uncertainty within the Southern Ocean, particularly for prokaryotes, could be attributed to the limited sampling conducted in this geographical region. This limitation resulted in a notable dissimilarity between satellite data collected in this area and the data sampled in the initial dataset, aligning with the findings of the reliability index. This finding is consistent with

**Table 3.** Results of the cross-validation and test exercises of SOMRCA and SOMChlF based on the regression coefficient ($R^2$), the root-mean-squared-error (RMSE), and the median relative deviation (MRD). One should note that, for SOMChlF, the $R^2$ and MRD were calculated using log-transformed Chla values and RMSE was calculated using real Chla values

| | SOMRCA Relative cell abundance (%) | | | | | | SOMChlF Phytoplankton chlorophyll-a fraction ($mg.m^{-3}$) | | | | | |
| --- | --- | --- | --- | --- | --- | --- | --- | --- | --- | --- | --- | --- |
| | Cross-validation n=115 | | | Test n=30 | | | Cross-validation n=115 | | | Test n=30 | | |
| | $R^2$ | RMSE (%) | MRD | $R^2$ | RMSE (%) | MRD | $R^2$ | RMSE ($mg.m^{-3}$) | MRD | $R^2$ | RMSE ($mg.m^{-3}$) | MRD |
| **Diatoms** | 0.65 | 2.7 | 0.74 | 0.72 | 2 | 0.58 | 0.66 | 0.24 | 0.18 | 0.86 | 0.12 | 0.20 |
| **Dinoflagellates** | 0.79 | 5.45 | 0.81 | 0.83 | 6.15 | 0.92 | 0.65 | 0.06 | 0.26 | 0.61 | 0.07 | 0.16 |
| **Green algae** | 0.61 | 5.64 | 0.61 | 0.67 | 4.32 | 0.47 | 0.71 | 0.08 | 0.20 | 0.62 | 0.19 | 0.23 |
| **Haptophytes** | 0.66 | 3.42 | 0.36 | 0.33 | 3.04 | 0.28 | 0.76 | 0.07 | 0.16 | 0.68 | 0.05 | 0.13 |
| **Prokaryotes** | 0.6 | 19.27 | 0.59 | 0.67 | 20.53 | 0.76 | 0.57 | 0.05 | 0.25 | 0.76 | 0.10 | 0.23 |
| **Cryptophytes** | 0.62 | 1.98 | 0.45 | 0.78 | 1.98 | 0.61 | 0.7 | 0.05 | 0.16 | 0.77 | 0.04 | 0.10 |
| **Pelagophytes** | 0.64 | 2.6 | 0.78 | 0.36 | 1.96 | 0.65 | 0.68 | 0.02 | 0.21 | 0.74 | 0.02 | 0.24 |
| **Chlorophyll-a** | | | | | | | 0.83 | 0.23 | 0.24 | 0.72 | 0.31 | 0.33 |

the documented very low abundance of cyanobacteria in the Southern Ocean (Flombaum et al., 2013), which may contribute to heightened model uncertainty for this particular region.

## 4.2 Comparison with global HPLC pigment dataset

The global in-situ HPLC dataset was then used to estimate Chla fractions for each phytoplankton group using the diagnostic pigment approach (DPA). This dataset was compared to the Chla fraction matching each phytoplankton group that was estimated by SOMChlF (Fig. 7). A total of 2671 matchups were found following the same procedure described in 3.1. Evaluating the sum of Chla fractions and comparing it with in-situ Chla can be considered as a baseline evaluation of this method. This comparison showed a satisfying correspondence score of $R^2$=0.72. Relatively good correspondence is noted for diatoms and haptophytes, showing an $R^2$=0.64 between in-situ and SOMChlF for diatoms and 0.65 for haptophytes. Moderate correspondence was found for green algae, cryptophytes, and pelagophytes, with an $R^2$ ranging between 0.43 and 0.39. Prokaryotes and dinoflagellates had the lowest correspondence between both outputs. The comparison between DPA-based phytoplankton groups and SOMChlF estimates is highly uncertain. It compares two types of information indicating the same phytoplankton group, with different underlying assumptions about how to define and describe a certain group. For some of the groups, these results are coherent. For example, the diatom Chla fraction is well captured by the latter, and the values agree with those estimated using HPLC observations; however, we noted a major overestimation within the HPLC DPA method. For prokaryotes, this comparison leads us to say that using zeaxanthin as an indicator of the cyanobacterial contribution to Chla may not entirely represent this group.

The permutation-based importance analysis using Random Forest, performed on the in-situ *Tara* Oceans *psbO* and HPLC measurements, emphasizes the necessity of a multivariate approach for predicting phytoplankton community structure based

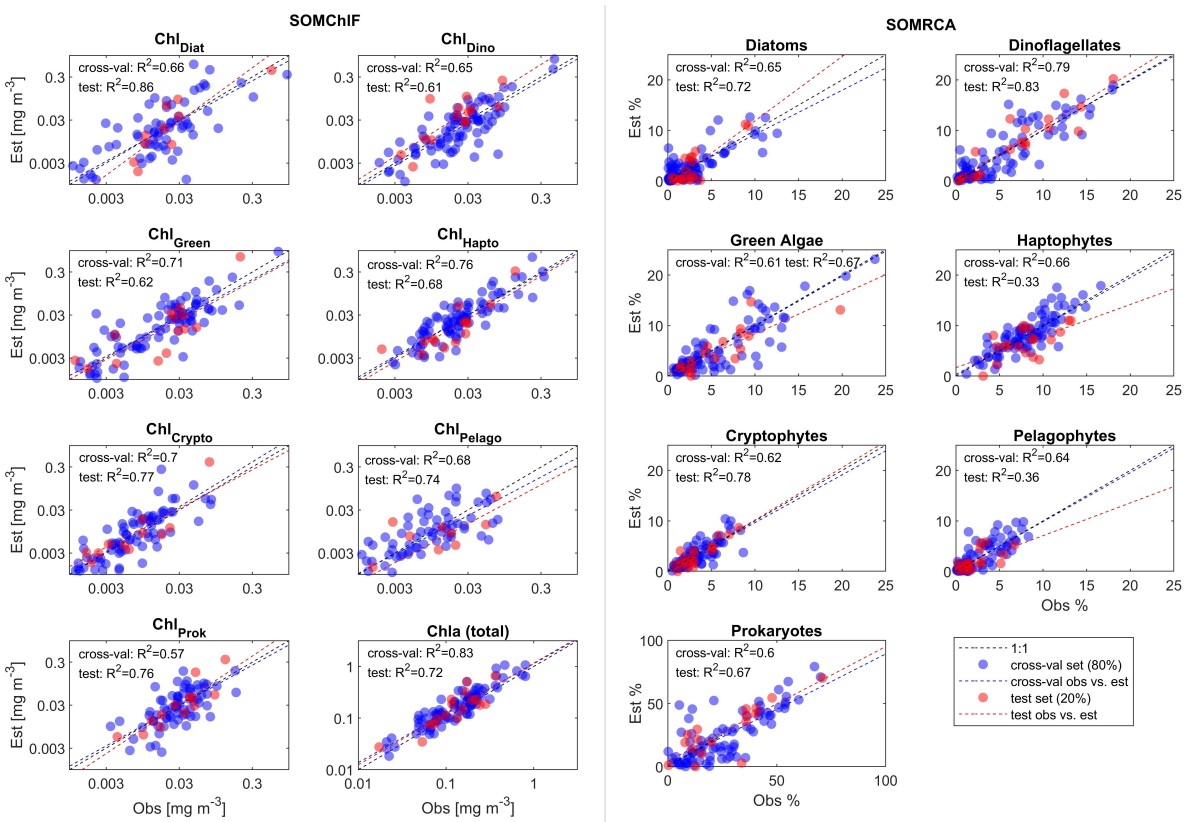

**Figure 5.** Results of the two-step cross-validation (blue) and test (red) procedures for SOMChlF (left) and SOMRCA (right) with the chosen best combination of satellite parameters and a SOM grid, respectively, of 242 and 222 neurons. From the initial dataset consisting of 145 observations, two sets were split: 80% to be used in a one-leave-out cross-validation procedure, and 20% as an independent test. For the cross-validation, each observation among the 115 observations, was used iteratively as a training set and as a test set until all observations served as tests (blue dots). This procedure was used to identify the best satellite combination and SOM grid size. Finally, the remaining 30 observations were used as a test to evaluate the generalization capacity of the SOM with the chosen configuration (red dots). One should note that, for SOMChlF, the $R^2$ was calculated using log-transformed Chla values. For complete evaluation metrics refer to Table 3.

on pigments (see Fig. 8). Notably, the diagnostic pigments mentioned in Table 1 exhibited dominant importance in determining the relative abundance of their respective assigned phytoplankton groups. For instance, peridinin represented dinoflagellates, Chlorophyll-b characterized green algae, and zeaxanthin indicated prokaryotes (Table 1). These pigments demonstrated the highest importance for their respective groups, as illustrated in Fig. 8, accompanied by a positive Spearman correlation. However, individually, these pigments accounted for less than 25% of the variance in their respective groups. Conversely, in the case of cryptophytes, diatoms, and haptophytes, no pigment stood out in terms of importance, and the observed correlations were related to co-variation between pigments (e.g., Chlb and Fuco in diatoms), possibly influenced by Chla variability. Therefore, the variability within each group is best explained not by a single diagnostic pigment, but rather by the overall pigment

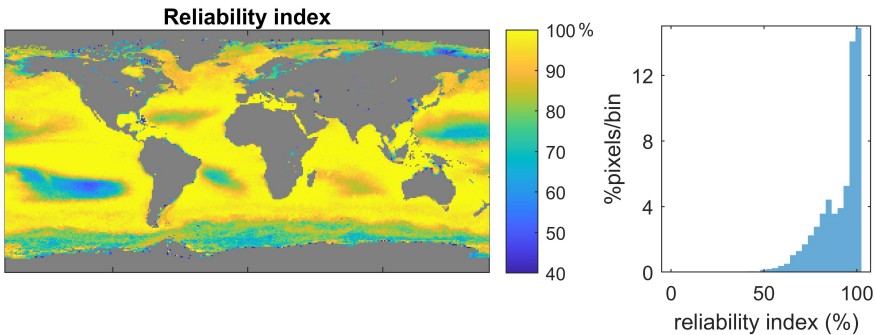

**Figure 6.** Applicability of the satellite *psbO*-based method. The geographical (left) and values distribution (right) of the reliability index were calculated between 1997 and 2021 by testing the set of satellite parameters at a given pixel against the values in the original dataset (D).

composition. It is crucial to consider how natural variability can influence the interpretation of pigment composition in relation to phytoplankton community structure. Pigment ratios not only vary with phytoplankton composition but also reflect the diverse strategies employed by different phytoplankton types to acclimate to environmental factors such as light, temperature, nutrients, and other variables.

### 4.3 Global patterns of satellite-derived phytoplankton groups

We then applied our method to Glocolour satellite data to generate a daily database spanning from 1997 to 2021, capturing the relative cell abundance and Chla fraction of seven phytoplankton groups of interest. Fig. 9 presents the annual patterns of relative cell abundance and Chla fraction for each phytoplankton group, derived from this satellite dataset.

Regarding relative cell abundance, the prokaryotes stand out as a dominant group. This group largely dominated tropical regions, with a relative abundance of up to 80% in subtropical gyres. Haptophytes, green algae, and diatoms exhibited higher 415 abundance in mid and high latitudes as well as the equatorial region, showing a maximum relative abundance of 30%. The remaining three phytoplankton groups displayed relative abundances that barely exceeded 10% of the total phytoplankton community. Pelagophytes and dinoflagellates were primarily observed in mid and subtropical latitudes, while cryptophytes were found in coastal areas and high latitudes.

Examination of how each phytoplankton group contributed to total Chla revealed that diatoms had a significant contribution 420 at high latitudes and equatorial regions. Prokaryotes, on the other hand, had an overall low to moderate contribution to total Chla.

Qualitatively, the information captured by SOMChlF was clustered into five groups, each characterized by a distinct remote sensing reflectance spectrum that corresponded to the phytoplankton community structure (Fig. 10). To illustrate the link between each group's contribution to total Chla concentration and relative cell abundance, we depicted the latter while evaluating 425 the pixel's assigned relative abundance values for each of the five clusters. This approach revealed that three out of the five clusters are dominated by prokaryotes in terms of cell abundance (C1, C2, and C3). However, based on their relative contribu-

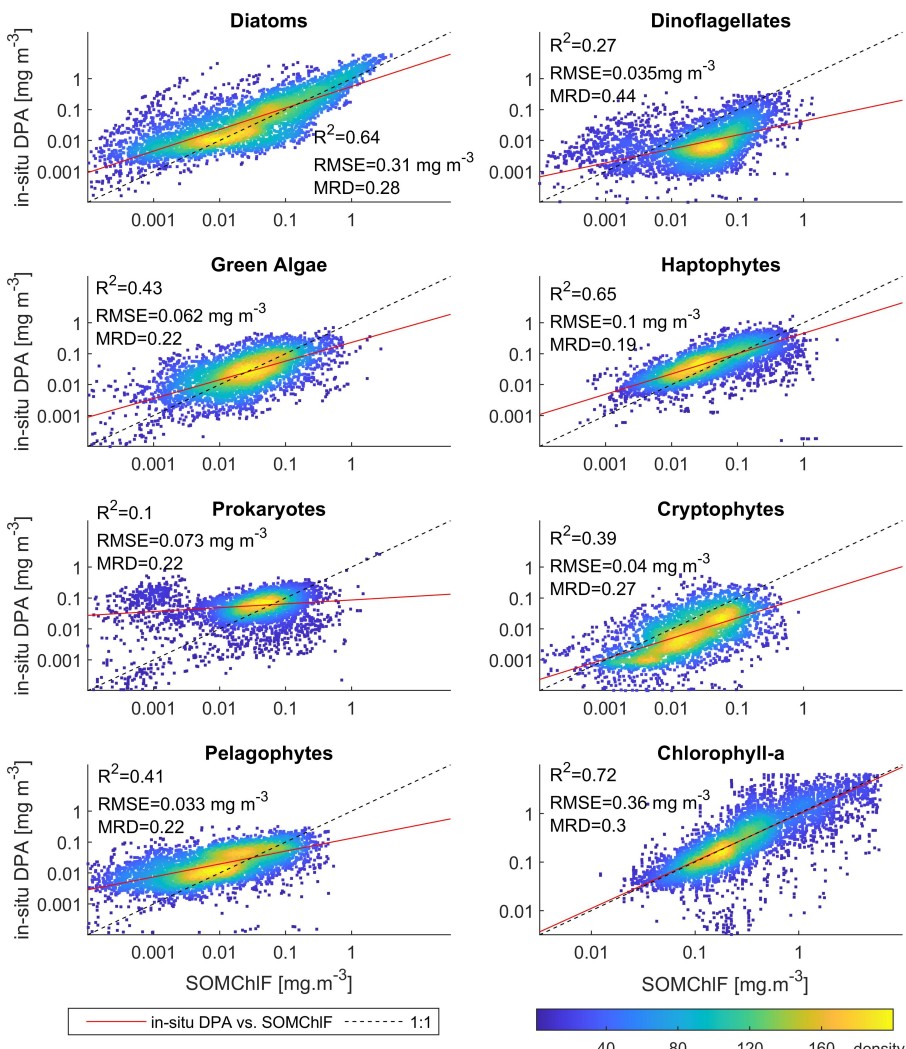

**Figure 7.** Comparison between the outputs of SOMChlF and the DPA approach applied on an in-situ global HPLC dataset. 2672 matchups were found between the outputs of SOMChlF and the in-situ dataset and analyzed in this figure. The $R^2$ and MRD result from calculations based on log-transformed data, and RMSE is based on non-log-transformed data.

tion to Chla, C1 was found to be dominated by prokaryotes and dinoflagellates, C2 exhibited a mixed composition, C3, and C4 represented diatoms and other eukaryotes, whereas C5 was predominantly composed of diatoms. The shift from relative cell abundance to size-integrated relative Chla fraction illustrates how cell size influences Chla contribution and variability.

Each cluster is characterized by a specific optical signature in terms of Rrs spectra. The Rrs values per wavelength were normalized based on their corresponding variance, enabling intercomparison regardless of magnitude. For instance, C1, which exhibits higher reflectance in the blue wavelength, represents clear, oligotrophic waters. In such environments with low nu-

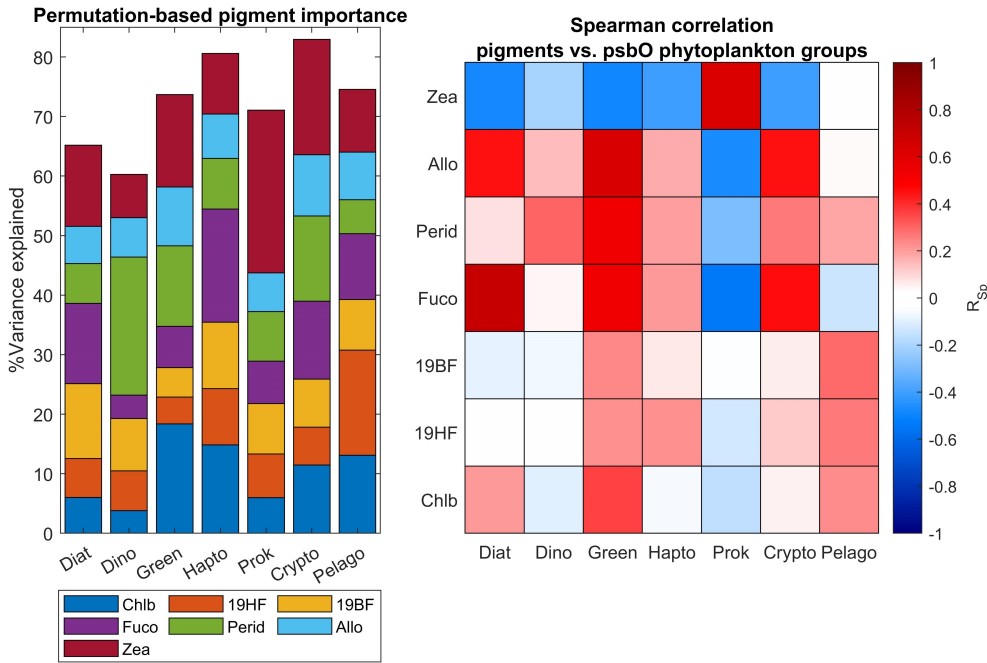

**Figure 8.** Evaluation of secondary pigment weighting for the estimation of different phytoplankton groups. The left panel represents the percentage of variance of each phytoplankton group explained by a set of frequently used phytoplankton secondary pigments. This analysis has been done using a random forest algorithm applied to the in-situ *Tara* Oceans *psbO* and HPLC datasets. A Spearman correlation coefficient has been calculated between each pigment and the phytoplankton groups (right panel).

trients and high surface stratification, picophytoplankton groups like cyanobacteria thrive due to their high surface-to-size ratio (Raven, 1998; Chisholm, 1992). C2 represents normalized Rrs spectra with insignificant differences between normalized bands, suggesting an average state where the phytoplankton community appears mixed. In C3 and C4, we observed an increase in normalized Rrs values in the green compared to the blue wavebands, indicating higher Chla in these environments. Given that C3 and C4 are located in high-latitude regions with ample nutrient resources and exceptional seasonal variability of light intensity, larger cell-sized phytoplankton groups, including diatoms, are favored, leading to increased biomass and Chla contribution (Brun et al., 2015). C5, with the greatest difference between Rrs in the blue and green, represents eutrophic waters, known for their high productivity and diatom-dominated blooms (Brun et al., 2015).

Based on the global distributions of these clusters, several biomes can be defined. C1 is centered in subtropical gyres, C2 is found in transitional zones such as mid-latitude regions and the equatorial region, C3 is observed in the Southern Ocean, C4 corresponds to high-latitude regions, and C5 is prevalent in coastal and eutrophic waters.

Different temporal variability is evident for each cluster across different latitudinal bands. In northern high latitudes, an increase in C5 indicates maximal productivity occurring in that region around May. At mid-latitudes, the winter maximum

is marked by an increase in C5 and C4 clusters. A secondary, less pronounced peak can be observed in autumn, attributed to the break in the thermocline and remineralization processes. During summer, C1 dominates the mid-latitude regions. In tropical regions, C1 is predominant, with a cyclic increase of C2 suggesting coastal influences, likely due to the proximity of C2 to nutrient-rich zones like upwelling systems. In contrast to northern high latitudes, the Southern Ocean exhibits a different temporal variability. The presence of prokaryotes is signified by C1 in this region, whereas C3 dominates during the bloom season in January. This analysis confirms the Antarctic nature of C3 in contrast to C4, highlighting differences in water types between the two regions based on phytoplankton community structure and satellite data.

This division into parallel and transitional biomes underscores the significant influence of latitudinal physical gradients, including light availability and temperature, on the structuring of the phytoplankton community in terms of types and size. These findings align with previous global phytoplankton studies conducted in situ (Ibarbalz et al., 2019; Sommeria-Klein et al., 2021) as well as satellite estimates (Alvain et al., 2006; Hirata et al., 2011; Ben Mustapha et al., 2013; El Hourany et al., 2019a; Xi et al., 2020, 2021).

## 4.4 Intercomparison of satellite-derived phytoplankton group products

A comparison was performed between SOMChlF's output and two operational products based on Xi et al., 2021 and SOM-Pigments (El Hourany et al., 2019a) algorithms. We based this on the five phytoplankton groups common to all three algorithms: diatoms, dinoflagellates, green algae, haptophytes, and prokaryotes for the year 2020. The annual patterns show a substantial agreement between all three satellite-derived phytoplankton estimates (Fig. 11). However, some differences between the estimated quantities of Chla phytoplankton groups can be noted. For diatoms, the outputs based on El Hourany et al. (2019a) and SOMChlF exhibit higher Chla values, while those based on Xi et al. (2021) show low values near the equatorial latitudes. For green algae and haptophytes, the three products show matching latitudinal variability, with only minor discrepancies in values at high and subtropical latitudes. For prokaryotes, the outputs of Xi et al. (2021) show higher estimates, particularly near the Arctic and equatorial regions. Lastly, for dinoflagellates, the SOM-Pigments method yielded lower Chla values, especially in subtropical gyres, whereas SOMChlF showed the highest Chla estimates for this taxonomic group.

Upon comparing the uncertainty patterns with those observed in Xi et al. (2021), similar trends were identified for the Chla fraction of eukaryotic phytoplankton, displaying consistency in following the Chla concentration gradient as seen in our study. Notably, regions such as the gyres exhibited lower uncertainties, whereas higher uncertainties were evident in high-latitude regions and marginal seas. Conversely, when examining the uncertainty in the retrieval of prokaryote Chla by Xi et al. (2021), lower uncertainties were noted in polar regions, contrasting with higher uncertainties observed in low-latitude regions. Similarly, in Brewin et al. (2017), the uncertainty maps for diatoms and dinoflagellates depicted distribution patterns akin to our uncertainty estimations in the North Atlantic Ocean.

This coherence in uncertainty patterns between HPLC-based products and our *psbO*-based product can be attributed to the direct relationship between DPA pigment concentration and total Chla, as well as between *psbO*-derived Chla fractions and total Chla. Consequently, similar patterns in predictions, as well as in the uncertainties, emerge.

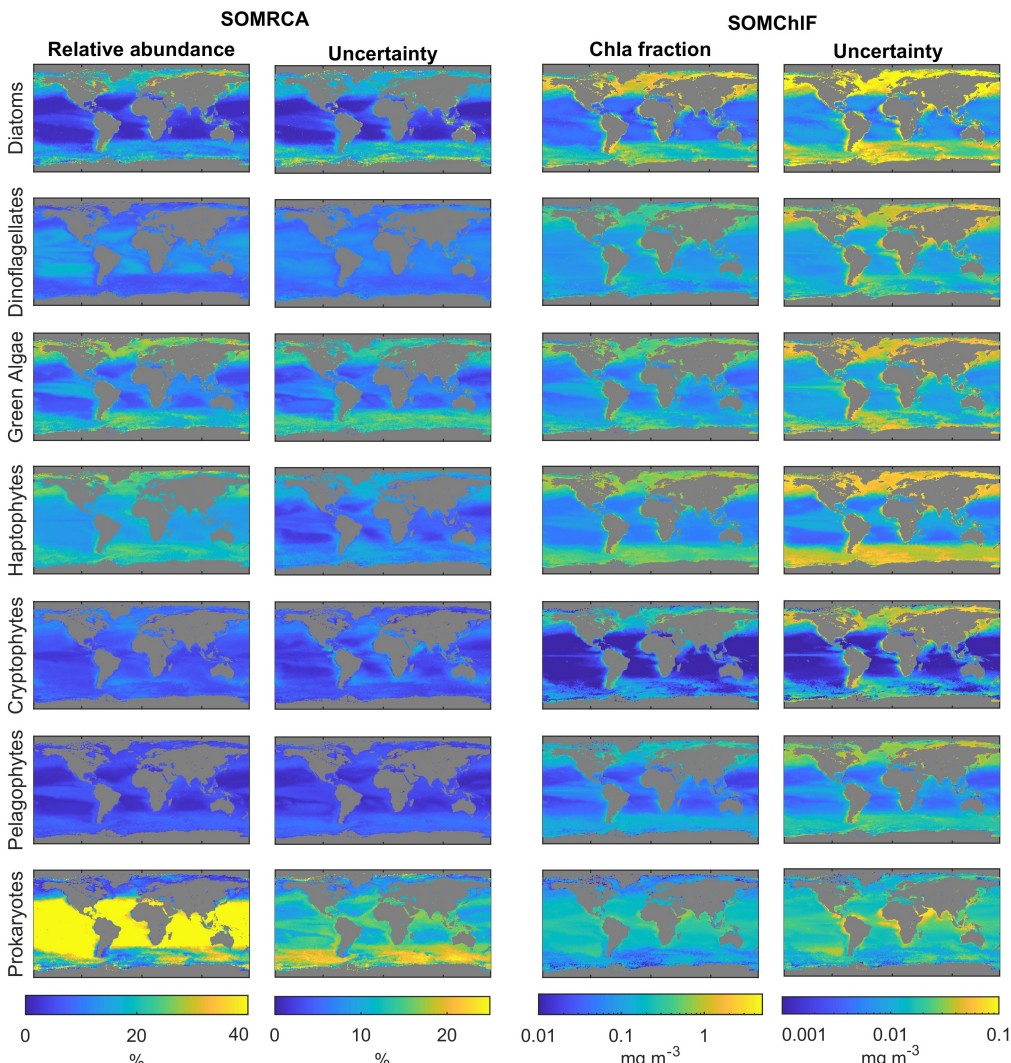

**Figure 9.** Annual composites of the relative abundances and Chl fractions of the seven *psbO*-derived phytoplankton groups based on satellite data (compiled using data from 1997-2021). The uncertainties related to each group and each method are because of their different possible combinations through the weighted standard deviations, as described in Section 3.2.3. We note that the scales for uncertainty are smaller than those in the abundance and Chla columns.

However, addressing the similarities and differences between the outputs of the above-cited methods referring to the same phytoplankton group is not a straightforward task. These methods are based on distinct assumptions and resolutions of phytoplankton groups; The estimation of phytoplankton groups using pigments is inherently imperfect and relies on assumptions that introduce considerable variability and bias in determining the contribution of specific pigments to the assessment of phytoplankton groups. For instance, several studies showed that the DPA approach tends to overestimate diatoms Brewin et al.

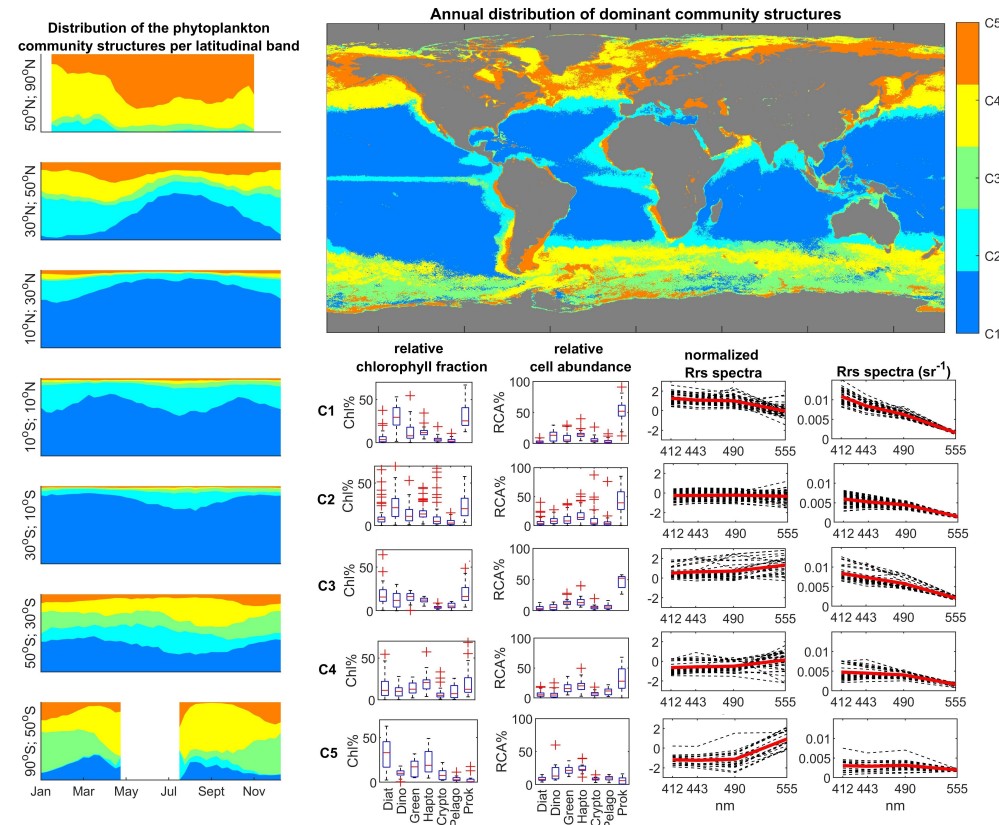

**Figure 10.** Satellite-derived biomes of phytoplankton communities, obtained by unsupervised clustering (Hierarchical clustering) of SOM-ChlF neurons. Relative cell abundances per phytoplankton group and normalized and denormalized Rrs spectra were also derived. The global map shows the most frequent community structure recorded during the 1997-2021 period. A spatio-temporal analysis was conducted to highlight latitudinal patterns.

(2014); Chase et al. (2020). This approach may compromise the relevance of satellite images when used. However, the added value of such an approach resides in the availability of the large HPLC dataset, which allows the development of robust algorithms. On the other hand, the method described in this paper and the generated outputs are based for the first time on a complete and harmonized database of phytoplankton taxonomic community structure on a global scale; an approach that provides an unbiased picture of phytoplankton cell abundances. However, the major limitation of this approach at this time is the low number of observations from which the metric has been derived.

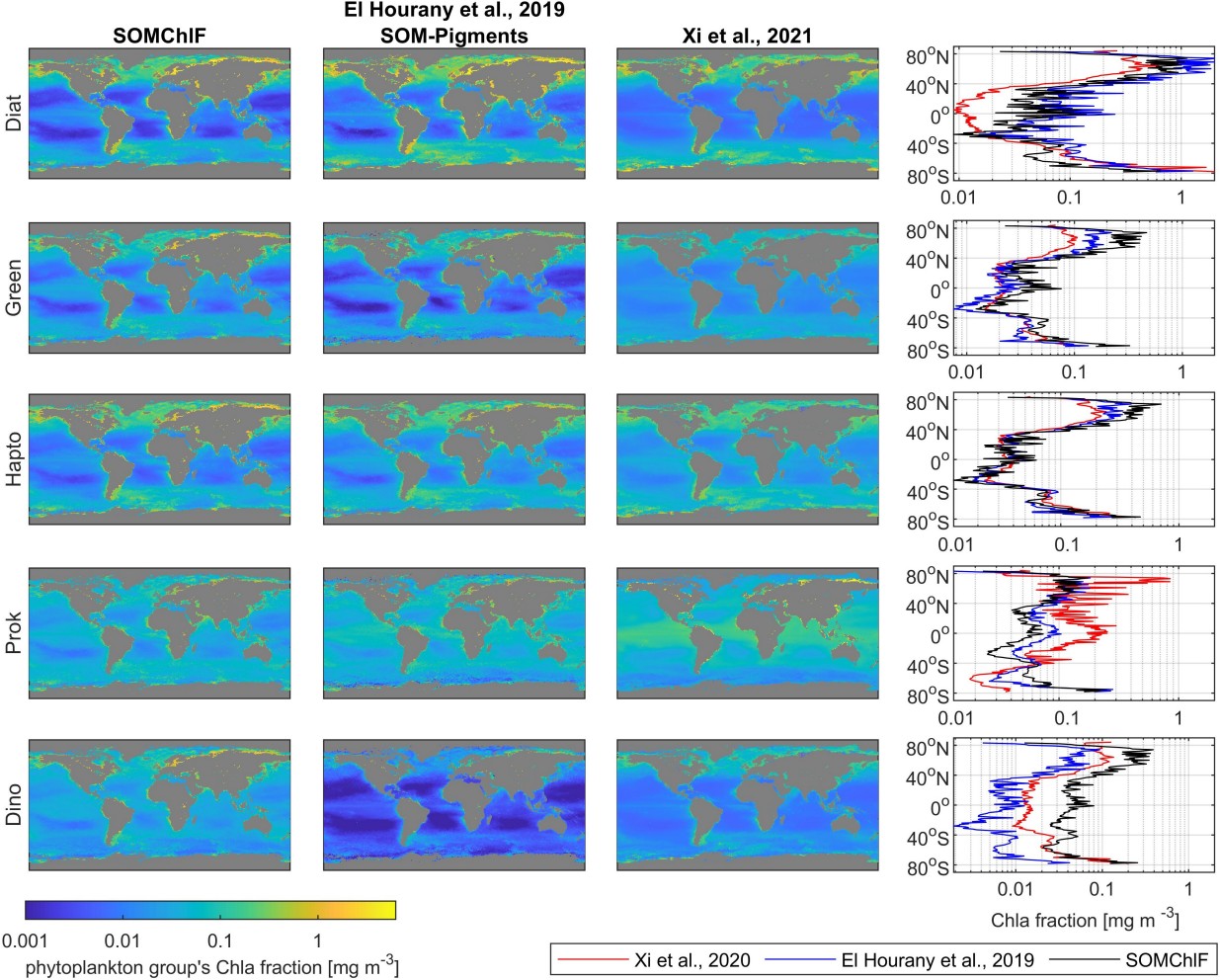

**Figure 11.** Intercomparison of five satellite-derived phytoplankton group Chla fractions based on SOMChlF, SOM-Pigments (El Hourany et al., 2019a), and Xi et al. (2021) algorithms for the year 2020. The annual average and the average per latitude of each Chla fraction are calculated to reveal global and latitudinal patterns.

## 5 Conclusions

By employing an alternative approach utilizing in-situ metagenomic observations, a reliable ocean color algorithm for detecting phytoplankton groups was developed in this work. This achievement is noteworthy considering the limited availability of omics data used in our analysis. The successful implementation was made possible by leveraging machine learning techniques and preserving the data structure using Self-Organizing Maps. The methodology demonstrated satisfactory performance in producing robust estimates for the seven major phytoplankton groups, albeit with some limitations in terms of global generalization due to the limited availability of data. For instance, it is important to exercise caution when interpreting estimates for

regions such as the subtropical gyres. As DNA sequencing costs continue to decrease and new expeditions generate molecular data from undersampled ocean regions, we expect the training datasets to increase rapidly in future years, which should further increase the accuracy of our method. Furthermore, this study presents a new global dataset of the relative cell abundances of the seven phytoplankton groups and their contributions to total Chla. These two types of information carry different implications. Chla serves as a biomass proxy, which is crucial for energy and matter fluxes in various ecological and biogeochemical processes. On the other hand, cell abundance represents species abundance for unicellular organisms, providing insights into community assembly processes.

This dataset opens up possibilities for inter-comparisons with existing approaches, such as DPA-based methods using in-situ and satellite data. The results provide coherent yet distinct information about phytoplankton communities, contributing to a better understanding of their composition. While our focus was on seven broad phytoplankton groups, it is worth mentioning that the deep taxonomic resolution achievable through molecular methods allows for species-level monitoring, which can be an interesting avenue for future implementation.

The methodology presented in this work provides a unique opportunity to observe in real-time and high-resolution the state of the major phytoplankton groups at the global scale. This makes remote sensing observations excellent tools to collect EBVs, play the role of broker between monitoring initiatives and decision-makers, and provide a foundation for developing marine biodiversity forecasts under different policy and management scenarios. To reach this objective, remote sensing data inherently needs to be validated with in-situ observations as well. Of further interest is the launch of NASA's Plankton, Aerosol, Cloud, ocean Ecosystem (PACE) mission, a strategic climate continuity mission that will make global hyperspectral ocean color measurements possible. This will allow extended data records on ocean ecology and global biogeochemistry, revolutionizing the detection of phytoplankton communities from space. From the perspective of the PACE mission, this study is a step towards further understanding the effect of environmental changes on phytoplankton community structure and diversity.

*Code and data availability.* *psbO* dataset: https://www.ebi.ac.uk/biostudies/studies/S-BSST761;

GlobColour dataset: https://www.globcolour.info/, https://hermes.acri.fr/.SSTCCIdataset: https://data.marine.copernicus.eu/product/SST_GLO_SST_L4_REP_OBSERVATIONS_010_024/description. Global HPLC pigment dataset: MAREDAT, POLERSTERN data, Labrador Sea expeditions data, and *Tara* Oceans Expedition data, all available on https://pangaea.de/, GeP&Co database (accessed at http://www.obs-vlfr.fr/proof/php/x_datalist.php?xxop=gepco&xxcamp=gepco), and finally the NOMAD: NASA bio-Optical Marine Algorithm Dataset, and the numerous campaigns found on the NASA SeaBASS portal were accessed at (https://seabass.gsfc.nasa.gov/). Following best practices, the two SOM algorithms will be deposited into a public domain repository accessible upon publication. Prerequisite software library SOM Toolbox 2.0 for Matlab is required, implementing the self-organizing map and Hierarchical Ascending Classification algorithm, Copyright (C) 1999 by Esa Alhoniemi, Johan Himberg, Jukka Parviainen, and Juha Vesanto and accessible at https://github.com/ilarinieminen/SOMToolbox. Matlab function for Random Forest algorithm was used to run the algorithm. MATLAB version R2020b, Statistics and Machine Learning Toolbox-Functions.

**Table 4.** List of acronyms

| Acronym | Definition | Acronym | Definition |
|---------|------------|---------|------------|
| PFT | Phytoplankton Functional Type | OLCI | Ocean and Land Colour Instrument |
| PG | Phytoplankton taxonomic group | PACE | Plankton, Aerosol, Cloud, ocean Ecosystem mission |
| PSC | Phytoplankton Size Class | IOCCG | International Ocean Colour Coordinating Group |
| HPLC | High Performance Liquid Chromathography | CCI | Climate Change initiative |
| DPA | Diagnostic Pigment Analysis | CMEMS | Copernicus Marine Environment Monitoring Service |
| DP | Diagnostic pigment | Rrs | Remote sensing reflectance |
| Chla | Chlorophyll-a | Kd490 | Attenuation coefficient at 490 nm |
| Fuco | Fucoxanthin | PAR | Photosynthetically available radiation |
| Perid | Peridinin | NFLH | Normalized fluorescence line hight |
| Allo | Alloxanthin | bbp | Particulate backscattering coefficient |
| Zea | Zeaxanthin | SST | Sea surface Temperature |
| Chlb | Chlorophyll-b | SOM | Self-Organizing Maps |
| 19HF | 19'-Hexanoyloxyfucoxanthin | AHC | Ascending Hierarchical Clustering |
| 19BF | 19'-Butanoyloxyfucoxanthin | TD | Truncated distance |
| DVChla | Divynil-Chlorophyll-a | D | Initial dataset |
| DVChlb | Divynil-Chlorophyll-b | DRCA | Phytoplankton groups' relative cell abundance sub-dataset to train SOMRCA |
| MAREDAT | MARine Ecosystem DATa | DChlF | Phytoplankton groups' chlorophyll-a fraction per group sub-dataset to train SOMChlF |
| PSII | Photosystem II | SOMRCA | SOM algorithm dedicated to estimating phytoplankton groups' relative cell abundance |
| OC | Ocean color | SOMChlF | SOM algorithm dedicated to estimating phytoplankton groups' Chla fraction |
| SeaWiFs | Sea-viewing Wide Field-of-view Sensor | RMSE | Root-mean-squared error |
| MODIS | Moderate Resolution Imaging Spectroradiometer | $R^2$ | Regression coefficient |
| VIIRS | Visible Infrared Imaging Radiometer Suite | MRD | Median relative deviation |
| MERIS | Medium Resolution Imaging Spectrometer | | |

*Author contributions.* Conceptualization, RE, ML, CB. Methodology, RE. Validation, RE, JPK. Formal analysis, RE, JPK, ML, CB. Investigation, RE, JPK, LZ, HL, ML, CB. Resources, ML, CB. Data curation, JPK, RE. Writing-original draft preparation, RE, Writing-review and editing, RE, JPK, LZ, HL, ML, CB. Visualization, RE Supervision, ML, CB Project administration, ML, CB. Funding acquisition, RE, ML, CB.

*Competing interests.* The contact author has declared that neither they nor their co-authors have any competing interests.

*Acknowledgements.* The authors acknowledge the recommendations and guidance of Emmanuel Boss (Pr. At the University of Maine) and Sylvie Thiria (Emeritus Pr. at the Sorbonne University). R.E. acknowledges CNES postdoc fellowship 2019-2021, ANR Junior professor (ANR-22-CPJ1), CNES TOSCA 2020-2021, Sorbonne University Emergence program 2021-2023, ML4BioChange. J.J.P.K. acknowledges postdoctoral funding from the Fonds Français pour l'Environnement Mondial. C.B. acknowledges ERC Advanced Award Diatomic (Grant agreement No. 835067), the Horizon Europe projects 'Marco-Bolo' (Grant Agreement No. 101082021) and 'BlueRemediomics' (Grant Agreement No. 101082304), and French Government 'Investissements d'Avenir' programs OCEANOMICS (ANR-11-BTBR-0008),

FRANCE GENOMIQUE (ANR-10-INBS-09-08), MEMO LIFE (ANR-10-LABX-54), and PSL Research University (ANR-11-IDEX-0001-02). This article is contribution number xxx of *Tara* Oceans.

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
