# Peer review of "Linking satellites to genes with machine learning to estimate phytoplankton community structure from space"

_EGUsphere, 2022_

## Author Response (AR1)

**General answer**

Dear Editors and Referees,

We would like to express our gratitude for the valuable comments and suggestions provided for improving our manuscript. We acknowledge the referee's observations regarding communication ambiguities and technical issues in the initial version, and we have prepared this revised manuscript to address these concerns and clarify the highlighted aspects.

This response aims to address the common major issues raised by both referees. We acknowledge that the development steps of the omic-based satellite algorithm in the paper were unclear, and the inclusion of a pigment-based approach for validation was misleading.

To clarify, our method is based on the link between omic and satellite data. The pigment approach only played a role in the post-training process to compare the outputs of both approaches. We incorporated the pigment HPLC data in this study due to its widespread use in ocean color remote sensing techniques for estimating phytoplankton groups, primarily because of its high data availability. However, it is important to note that numerous studies have demonstrated significant uncertainties between the pigment approach and phytoplankton abundance observed through other methods (Chase et al., 2020). These uncertainties arise from factors such as the overlapping presence of pigments across phytoplankton classes, photoacclimation, and physiological processes. Therefore, it is crucial to recognize that our study addresses two types and levels of information: Omics and Pigments. The use of pigments in this work is more for comparison purposes rather than validation, and we acknowledge that our previous message regarding this matter was misleading.

Additionally, the referees found it unconvincing to introduce physiological uncertainties when transforming omics data into Chla fraction per phytoplankton class. We introduced this aspect to compare the omic and pigment-based approaches.

Based on the comments from both referees, we have chosen to thoroughly revise the methodology section. We have added flowcharts to simplify the process and enhance its applicability for readers. The entire methodology has been revised in light of the suggestions provided by the referees. To address the concerns regarding chlorophyll-a fractionation and enable the emergence of different levels of information as outputs, we trained two algorithms using the same satellite data and SOM methodology. One algorithm provides the relative cell abundance of phytoplankton; SOMRCA (including the estimation of direct psbO relative abundance values), while the other algorithm estimates the phytoplankton Chla fraction per group; SOMChlF. Importantly, this revised methodology now considers the psbO occurrence per size fraction, which was not taken into account in the initial version of the manuscript. In both algorithms, uncertainties on the outputs were evaluated and therefore are presented with the outputs.

[Figure]

The outputs from both algorithms will allow us to address questions regarding phytoplankton diversity from an ecological perspective (through relative cell abundance) and a biogeochemical perspective (through Chla fraction per group), while considering physiological uncertainties.

We sincerely hope that this response is convincing and meets your expectations. We appreciate the thorough review process and are confident that the revisions have significantly improved the manuscript.

Figure 1: Different levels of information on phytoplankton groups. Noting that cell relative abundance (SOMRCA) and Chla fraction per group (SOMChlF) are two outputs of two different algorithms based on the same SOM methodology

**Comments Referee #1**

The work by El Hourany et al. describes machine learning techniques for application to (blue) ocean color data to determine the global distribution of phytoplankton functional types. A special focus is on the description of ML techniques with the identification of crucial features based on parameters of the merged GlobColour dataset. The details of the methods used are often cryptically written and difficult to follow, and reproduction of the methods and results is not possible. The methods section should be revised accordingly. Besides the application of ML methods in the context, the advantage of the method remains unclear and is not further specified; it could well be higher accuracy or computing speed. I recommend a thorough revision of the paper to describe the methods in a more understandable way and to prove the added value (also of future ML methods).

We thank the referee for the valuable comments and suggestions provided for improving our manuscript. We acknowledge the referee's observations regarding communication ambiguities and technical issues in the initial version, and we have prepared this revised manuscript to address these concerns and clarify the highlighted aspects.

In this paper, we approach the estimation of phytoplankton groups as a unified community structure to preserve inter-group coherence. Our objective was to develop a method capable of estimating all seven groups using a single set of satellite predictors. The challenge we faced was twofold: the problem was multivariate in nature, and the dataset was relatively small, with missing values in the satellite matchups.

To address these challenges and ensure that no valuable psbO measurements were lost, we turned to the technique of Self-Organizing Maps (SOM) and topology conservation. SOM is a powerful unsupervised learning algorithm that allows for the establishment and reproduction of relationships between variables. By utilizing SOM, we were able to fill in the gaps in the dataset and exploit the preserved topology to estimate the phytoplankton groups.

The advantage of using SOM in this context is its ability to handle multivariate data and preserve the underlying structure of the variables. It enables us to capture the complex relationships between the predictors and the phytoplankton groups, even with missing values. By leveraging the topology conservation property of SOM, we ensure that the estimated relationships are consistent with the overall structure of the data.

Specific comments:

The title is a bit catchy and inaccurate. It is rather about pigments, which are typical for color groups, but which can be very different in type of phytoplankton and corresponding genes.

Indeed, in this work, pigments were used, but not for training the method. We appreciate the referee's concerns regarding clarity, and we would like to address them.

The method we introduce in this manuscript is based on a dataset of phytoplankton groups quantified using the psbO molecular method and expressed in terms of Chla fraction, in combination with satellite variables. As described in the text, psbO is a single-copy gene that is present across all phytoplankton groups. This gene encodes proteins that structure a compartment of the Chloroplast photosystems.

It is important to note that pigments were not used for training the SOM method. Instead, they were employed solely for comparative purposes. The outputs of SOM-psbO (previous version of the algorithm) were compared to in-situ phytoplankton groups estimated using diagnostic pigment analysis (DPA), as described in studies such as Soppa et al. (2014). DPA methods have been widely used in remote sensing studies to estimate phytoplankton functional types (PFT) or size classes, and current operational methods such as Xi et al. (2020) and PHYSAT are based on them. However, it is crucial to acknowledge the high uncertainties associated with DPA methods, as highlighted by Chase et al. (2020). These uncertainties can lead to misleading interpretations of real PFT and phytoplankton size class relative abundance.

Therefore, it is important to clarify that diagnostic pigment data were not utilized in the development of the SOM method in both versions of the algorithm (old and revised). Their inclusion was solely for comparison purposes, to highlight the differences and uncertainties associated with the pigment-based approach.

We hope this clarification addresses the concerns regarding the use of pigments in our study.

The figures should all be revised, e.g. Fig. 5. Axis labels with units are often missing. Partly chlorophyll concentrations are given in log10, this is better in Fig. 2.

All figures were revised according to the referee's suggestion.

Line 87: Only as a comment that size fractioning often damages the cells, and such data should therefore be treated with caution.

We are aware of the drawbacks of size fractionation. The filters may retain cells smaller than the nominal pore because of net clogging, or because they were trapped in fecal pellets. On the contrary, long needle-like species and broken cells and colonies can pass through small mesh sizes. The patterns that we described in the current work based on size-fractionated samples can be complemented in the future by exploring non-fractionated samples. However, there is still no equivalent standardized sampling covering the main ocean regions as the size-fractionated samples from Tara Oceans.

For the discrimination of absorption features, rather the central visible region is necessary (e.g. Xi et al. 2015). In this respect, the use of the GlobColour data set with Rrs only up to 555 nm is unfavorable, as the correlation plots show. The OC-CCI dataset has more (MERIS) bands here and corresponding differences could be underlined. References to GlobColour and matchup procedure are missing.

As the referee correctly pointed out, the SOM-psbO method was trained using a dataset that included 17 variables, including satellite reflectance at 412, 443, 490, and 555nm. We apologize for not clearly indicating in the initial version of the manuscript that this method is specifically developed for open ocean applications.

In the clear open ocean, beyond 555nm, the information contained in the remote sensing reflectance (Rrs) bands is limited due to the strong absorption by water, as also mentioned in Xi et al. (2015). Our choice of the range and number of satellite reflectance bands was inspired by the work conducted in the PHYSAT method (Alvain 2005, Ben Mustapha et al., 2013), which is a classification method that utilizes reflectance anomalies in the four selected bands to identify dominant phytoplankton functional types.

To further support our argument, we rebuilt and cross-validated our methodology using different combinations of 15 bands ranging from 412 to 709nm. However, we found that increasing the number of bands did not lead to a significant improvement in performance. It should be noted that the Rrs bands selected, including the additional 670 nm band, are commonly measured by all sensors used to build the Rrs product of Globcolour. This overlapping of different sensors enhances data availability and coverage, thus increasing the importance of these Rrs bands within the initial dataset. The inclusion of the Rrs at 670 nm did not significantly impact the performance of either SOMRCA or SOMChlF, primarily due to the open ocean nature of the dataset.

It is important to note that one of the advantages of using machine learning methods such as SOM is to reduce the complexity of the problem while capturing non-linear relationships that are present in the environment. The correlations with the Rrs bands, which the referee mentioned as unfavorable in Figure 6, are indeed essential and statistically significant. It is crucial to consider that the problem we are addressing is multivariate. Preserving the inter-variable relationships, even those with lower correlations, is a major advantage of utilizing such a machine learning method.

It is a Case-1 approach for a medium range of chlorophyll concentrations, which should be communicated in a better way. Maybe flagging and an uncertainty product would be useful. Indeed, as clarified in the previous comment, the method developed in this paper is specifically designed for open ocean (case 1) applications. This statement has been further clarified in the revised version of the manuscript.

The approach proposed in this paper to estimate phytoplankton groups from satellite data is based on an unsupervised neural classification technique, specifically the Self-Organizing Map (SOM). The SOM summarizes the non-linear relationship between the satellite data and phytoplankton groups, effectively reducing noise and mitigating the influence of uncertainties within the dataset.

The function that links the predictors (satellite data) to the predicted variables (phytoplankton groups) is represented by an allocation function based on a weighted Euclidean distance. In other

words, this function searches for and associates the closest neuron in the SOM to a new or unfamiliar observation.

The main source of uncertainties in the estimation process lies in the allocation function. Among hundreds of neurons in the SOM, one neuron is chosen as the assignment based on the minimum distance between the neurons and the pixel, regardless of whether the distance is strong or weak. Since one of the properties of SOM is the preservation of topology (where neighboring neurons are similar), a pixel can be assigned to several adjacent neurons, with a distance order, representing a neighborhood of close neurons.

Now, how do uncertainties in the satellite variables influence the allocation function and, consequently, the results?

If the distance between a pixel and a neuron is small, the influence of uncertainties is minimal and will not significantly affect the assignment of the pixel. However, if a large distance is observed between the observation and the assigned neuron, uncertainties in the variables can have a greater impact on the choice but remain within the bounds of the chosen neuron's neighborhood.

To consider all the uncertainties associated with the allocation function, we have chosen to associate each pixel with a weighted standard deviation based on the first 10 closest neurons. The weights correspond to the distances between the first 10 matching neurons and the pixel. This allows us to incorporate uncertainties into the assignment process and provide a measure of confidence for each pixel's assignment.

By considering the weighted standard deviation, we account for the influence of uncertainties in the satellite variables and provide a more comprehensive understanding of the allocation process within the SOM.

Figure 2: Global uncertainties regarding phytoplankton groups' cell relative abundance, Chla contribution (SOMRCA), and Chla fraction (SOMChlF). In this context, the following uncertainties on the outputs represent the interval (defined with a standard deviation calculated on the neighboring associated neurons per satellite pixel) of SOMRCA and SOMChlF to estimate the different phytoplankton groups.

However, in such open ocean conditions, HPLC methods are often at the limit (if low volumes of water are filtered) – extreme uncertainties may exist in the fundamental training data.

The very deep sequencing of the *Tara* Oceans metagenomes (between ~10^8 and ~10^9 total reads per sample) allows high detection power (e.g., for rare species). In addition, filter volumes were high: 100 L for 0.22-3, 0.8-5 and 5-20, 1-20 m3 for 20-180, and 10-100 m3 for 180-2000.

Besides SST is salinity actually a strong indicator for some PFTs.

SSS is a strong indicator of some PFTs due to intervariable correlations, and their patterns are related to physical conditions, like the ones of SST. However, SSS satellite products are not as accurate as SST products and at a lower resolution (best resolution at 25kms vs. 4kms). The addition of Satellite SSS products might corrupt the output of the operational phase.

Method part is unclear, especially lines 163-212. A part of the problem could be that less common naming conventions are used, e.g. do you refer to neural network architecture if you optimize the size map? How does the final map or architecture look like?

We acknowledge the referee for highlighting these communication issues. We introduced a clearer definition of the SOM size; We refer to the number of neurons represented by n=p x q, where p and q are the dimensions of the SOM 2D neuron grid.

Line 269: The more parameters we utilize, the more we must trust the data quality. Nevertheless, seen over the global ocean, there are many uncertainties in all mentioned parameters and regions. Especially Rrs in blue bands and the retrieved chlorophyll concentration must be considered as critical, even more because reflectances are derived from multi-mission merged data with sensor-specific atmospheric correction.

The question raised highlights the importance of considering data quality when utilizing parameters. In the context of the global ocean, there exist numerous uncertainties associated with the mentioned parameters and regions. As mentioned in the previous comment regarding uncertainties, the SOM process attunes uncertainties and enables the possibility to estimate uncertainties in the outputs. This has been implemented in the second version of our algorithm.

The marine model of ocean color algorithms is for atmospheric correction and chlorophyll retrieval is mostly based on a diatom-like chlorophyll-specific absorption and scattering behavior (e.g. Bricaud et al., 1995). Thus, good that there is relatively high correlation of diatoms and chlorophyll concentration. But what is actually with features that are not captured, e.g. specific optical properties of Coccolithophores (e.g. Balch, 2018)? There is a high abundance, e.g. in The Great Calcite Belt, where Fig. 7 indicates high reliability of the model with a C2 distribution in Fig. 10, that seems to be different. I see some question marks and would ask for more careful discussion about the model uncertainty.

We admit that within the first version of the algorithm, since we didn't take into consideration the effect of size per group and per sample, the Chla fraction concentration per group was biased. The pos-training classification (Figure 12 in the revised manuscript, section 4.3) into dominant phytoplankton communities was revised accordingly after incorporating the phytoplankton size information as described in Sommeria-Klein et al 2021 Science:

$$Chla\ fraction_{PFT} = \text{Chla}_{in-situ} * \frac{\sum_{s=1}^{4}\left(\frac{psbO_{PFT} * size_s}{\sum_{PFT=1}^{7}(psbO_{PFT} * size_s)}\right)}{\sum_{s=1}^{4}\sum_{PFT=1}^{7}(psbO_{PFT} * size_s)}$$

Therefore, when converting psbO reads to relative abundance, considering the size of the phytoplankton cell for each group, we highlight the contribution of each group's size to the total chlorophyll-a (Chla) concentration.

Compared to the previous version, and due to the data conversion, five clusters turned out to be sufficient to describe the dominant patterns. In the Southern Ocean, the C3 group emerges and dominates, while there is also a higher relative abundance of Haptophytes and Diatoms. In the Arctic Ocean, the C4 group dominates. Although the phytoplankton communities of both C3 and C4 clusters were relatively similar, the optical signal was significantly different, allowing us to distinguish between the two clusters.

It is unclear how the new method behaves compared to the mentioned operational model by Xi et al. (2020). What are the advantages of the presented method?

Xi et al., (and the SOM-Pigments method) is based on the DPA pigment approach to identify phytoplankton groups (4 functional types).

The method described in this paper is developed with a harmonized database on the phytoplankton taxonomic community structure based on the psbO gene quantification. Molecular methods like this have a deep taxonomic resolution (including for cryptic species) as well as high detection power (e.g., for rare species). In addition, this particular gene is present in all phytoplankton groups, eukaryotes, and prokaryotes alike, with a single copy per cell.

Quantifying it using satellite data provides an unbiased picture of phytoplankton cell relative abundances.

**Comments Referee #2**

The authors develop a machine learning approach to link ocean colour data and in situ omics to improve detection of phytoplankton functional types and groups from space. The topic they are dealing with is innovative. However, the methodology and algorithm development steps are hard to follow and need to be revised to make the workflow clearer to the reader. In this scope, a flowchart is essential.

I am not fully convinced by the validation approach of the method. The training is done using the whole omics database and cross-validation statistics show the good prediction capabilities of the model. Then, the validation is made with an external database built on HPLC-based information. From my point of view, this cannot be considered a proper validation because one quantity is based on HPLC data, the estimated one on omics data. Such a comparison thus implies that the two approaches bring the same level of information on phytoplankton taxonomy. In this case, there would be no need to develop a new approach based on omics. However, as discussed at the end of the paper, HPLC- and omics-based phytoplankton information have some degree of correlation, which is good because this means that OMICS information can be found in optical properties to some extent and OMICS based approaches are welcome because they will bring new and complementary information on phytoplankton from space.

I realize that the OMICS database used to develop the new ML approach is small, but probably the authors might think to train the model over 70% of the database and validate it with the remaining 30%.

Results need to be discussed more and the text about retrieved global distribution of phytoplankton and biomes needs to be profoundly checked and revised.

The work thus needs to be deeply revised to improve the methodology and make the validation stronger as well as the text more readable.

We would like to express our gratitude for the valuable review. We acknowledge the referee's observations regarding technical issues in the initial version, and we have prepared this revised version while applying the referee's suggestions.

We would like to admit that the reasoning behind validating with a pigment-based approach was misleading. For that, we chose to follow the referee's major comment and evaluate the algorithm using a two-step procedure:

We split the Tara Oceans psbO dataset into 80% to train the SOM, and 20% as a test set.

1-    During the SOM training based on 80% of the dataset, a different combination of satellite variables was used to determine the best set of variables to estimate the 7 phytoplankton groups in terms of relative cell abundance and Chla fraction.

·        Per a combination of variables, we increase the number of neurons to determine the optimal size of the SOM from 10 neurons to 1000 neurons.

o   For each number of neurons used, the quantization and topographic errors related to the SOM are calculated and a one leave-out cross-validation procedure is performed to assign performance metrics (R2 and RMSE) to help choose the best SOM size and satellite variables combination.

The best SOM configuration and variable combination are based on the lowest errors and highest R2 values.

2-      The chosen SOM is tested using the 20% test set, providing an independent set of performance metrics.

As a result, we present in the paper the performance metrics of the best SOM configuration based on the cross-validation procedure and the test set.

The comparison with the HPLC DPA approach will be introduced for comparative purposes only.

Specific comments:

Figure 1 is misleading as the same color palette has been used for both columns though the % axis are different from left to right. A quick reader could interpret the yellow dots of (e.g.) Cryptophytes as abundant as Green Algae or Diatoms.

Indeed, according to the referee's comment, we homogenized the color scale. And since we derive two types of information from the psbO dataset i.e relative cell abundance and Chla fraction per group, a different color palette was used for each new dataset.

Line 91: this statement means that we have phytoplankton also in the 180-2000 um size class, which is possible in case of diatoms chains. Could you provide a distribution of frequency of phytoplankton groups within each size class? This would help the reader to have a wider image of the type of phytoplankton in the database (and especially for those chain-forming species and classes spanning a wide size range).

The distribution of taxonomic groups between size fractions in the psbO dataset is displayed in Fig 2a-b, Fig 7a, Fig 8 and Figure S16 in Pierella Karlusich et al 2023 Mol Ecol Res.

We provide boxplots to illustrate the distribution of the phytoplankton groups per size filter.

The filters may retain cells smaller than the nominal pore because of net clogging, or because they were trapped in fecal pellets. On the contrary, long needle-like species and broken cells and colonies can pass through small mesh sizes. The patterns that we described in the current work based on size-fractionated samples can be complemented in the future by exploring non-fractionated samples.

Line 115: why normalizing omics data on Chl? Because Chl varies according to the physiological status of phytoplankton, a photoacclimation component is re-introduced (which is a major problem in the DPA analysis). Why not using OMICS-based % of the whole population?

Indeed, this type of normalization introduces physiological uncertainties in the data. However, it was judged important to achieve quantity relative to Chla which is often used as a proxy of biomass, and which is a relevant parameter for energy and matter fluxes (e.g., food webs, biogeochemical cycles). Adding to this, this Chla normalization allows to compare this quantity with what we can observe using DPA pigments approach and current satellite operational products.

But, as mentioned in the general answer two types of algorithms are developed to address this issue:

**To address the concerns regarding chlorophyll-a fractionation and enable the emergence of different levels of information as outputs, we trained two algorithms using the same satellite data and SOM methodology. One algorithm provides the relative cell abundance of phytoplankton (including the estimation of direct psbO relative abundance values), while the other algorithm estimates the phytoplankton Chla fraction per group. Importantly, this revised methodology now considers the psbO occurrence per size fraction, which was not taken into account in the initial version of the manuscript. In both algorithms, uncertainties on the outputs were evaluated and therefore are presented with the outputs.

The outputs from both algorithms will allow us to address questions regarding phytoplankton diversity from an ecological perspective (through relative cell abundance) and a biogeochemical perspective (through Chla fraction per group), while considering physiological uncertainties. **

Lines 121-123: it is not clear which data are interpolated. In situ or satellites?

In-situ and Satellite data are both interpolated within the initial data space during the training phase, since missing values can be present in both cased, and since the number of neurons that we are dealing with is greater than the amount of observation. This is monitored through the training process with both the quantization and topographic errors related to SOM.

Table 2 contains mistakes on the coefficients. From Uitz et al. (2006), the coefficient for Chl-b is 1.01 while 0.35 is for 19-BF. Fixed, we apologize for this mistake.

In addition, 19-BF is here only attributed to the pelagophytes while is also a pigment within haptophytes (except coccolithophores). So, from the current coefficients all haptophytes only contain 19-Hex.

Indeed, in the study by Chase et al. (2020), it was demonstrated that the presence of pigments overlaps within size classes and types of phytoplankton. Several ocean color studies, such as those by Hirata et al. (2011) and Xi et al. (2020), have attributed the 19Hf pigment to Haptophytes.

Taking into account the reviewer's comments and the uncertainties associated with pigments, we decided to list the major phytoplankton groups and indicate the most representative pigment for each group.

Line 155: which cross-validation procedure? Do these statistics refer to all pigments or is it a global indicator for the technique?

We acknowledge that the sentence citing the statistics was unclear. The reported statistics, a regression coefficient of 0.75, and an average RMSE of 0.016 mg.m-3, represent a global indicator for the technique. They reflect the mean error and regression coefficient across the 10 estimated pigments and are given following a cross validation procedure conducted using a one-leave-out random pick from a global HPLC dataset constituted of 12 000 HPLC observations.

Line 163: Please indicate and explain better which are the "several machine learning algorithms" you tested and why a SOM has been chosen. This will be very helpful for scientists approaching the same problem.

We would like to clarify the sentence introducing the machine learning algorithms used in our study: SOM, hierarchical ascending clustering (HAC), and Random Forest.

Developing an operational algorithm that estimates the abundance of phytoplankton groups from satellite information was achieved using these algorithms. Firstly, the SOM algorithm was utilized to train a model based on the psbO pigment dataset. This allowed us to identify global large-scale patterns and characterize phytoplankton biomes. Last, to explain the potential divergence between the DPA approach and psbO measurements, we employed a Random Forest approach. This analysis highlighted the cumulative importance of pigment composition in estimating the abundance of phytoplankton groups.

We tested approaches based on Feed-Forward Neural Networks. However, due to the limited number of observations in the dataset, these approaches were not very conclusive. The choice of SOM was based on the previous work by El Hourany et al. (2019), which demonstrated improved performance with an increasing number of neurons, the number of neurons almost twice compared to the observations in the initial dataset, accounting for missing values.

In the following section, each methodology and algorithm are explained in detail. Section 3.1 need to be rewritten and a flowchart added. That's strange to see 3.1.1 and 3.1.2 as two different sections when (if I had well understood) the work is done simultaneously. Figure 5: y- and x- axes should be the same and indicate the name of the solid and dashed lines in the caption.

We apologize for the misleading sectioning. To better clarify the methodology, a flowchart was added and both above-mentioned sections were merged according to the methodology as the reviewer mentioned. Indeed sections 3.1.1 and 3.1.2 are done simultaneously, and iteratively as shown in the new flowchart #2.

[Figure]

Flowchart 1: General scheme of the SOM methodology to estimate phytoplankton groups from satellite data.

**Training procedure**

Flowchart 2: A focus on the training phase of the SOM which is based on an iterative procedure between different satellite variable combinations and SOM grid size. The choice of the best satellite variables combination and SOM size were based on consensus of low errors and high R2.

Line 191: which several experiments? How many? Please explain better.

The SOM grid size was sampled between 10 to 1000 neurons with a step of 10. Therefore, there were 100 SOM grids that were tested for each variable combination.

Section 3.1.3 needs to be clearer.

Line 269: what is the impact of interpolation on bbp and Kd? (i.e., Interpolation declared in the methods)

Below is a comparison of SOM-psbO and the initial dataset's values for each variable including bbp and Kd. For a SOM grid size of 242 neurons, the SOM was able to catch the values' distribution for both parameters.

[Figure]

Figure 3: Distribution of values for each variable in the initial dataset and SOMRCA and SOMChlF neurons.

Line 275: from Table 3, pelagophytes instead of cryptophytes

Indeed, we apologize for this error.

Line 314: generally speaking, are you referring to the surface-to-volume ratio?

We have corrected the term: 'biovolume-to-size' was replaced by 'surface-to-size.'

Line 329 and Line 331: please check and discuss: C4, C5 and C6 are dominated by Prokaryotes, but these areas are generally known to be dominated by large phytoplankton. Same for C1, dominated by diatoms but in the subtropics. In addition, it would be nice to see these clusters plotted on map in Figure 10.

We admit that within the first version of the algorithm, since we didn't take into consideration the effect of size per group and per sample, the Chla fraction concentration per group was biased.

The pos-training classification into dominant phytoplankton communities was revised accordingly after incorporating the phytoplankton size information as described in Sommeria-Klein et al 2021 Science:

$$Chla\ fraction_{PFT} = \text{Chla}_{in-situ} * \frac{\sum_{S=1}^{4}\left(\frac{psbO_{PFT} * size_s}{\sum_{PFT=1}^{7}(psbO_{PFT} * size_s)}\right)}{\sum_{S=1}^{4}\sum_{PFT=1}^{7}(psbO_{PFT} * size_s)}$$

Therefore, upon converting psbO reads to relative abundance accounting for the size of the phytoplankton cell per group, we highlight the size contribution of each group to the total Chla.

Compared to the previous version, and due to the data conversion, five clusters turned out to be sufficient to describe the dominant patterns (Figure 4).

Figure 10: How the spectra have been normalized? By the minimum? The spectral shape should be discussed.

Each wavelength was normalized by its values distribution variance within the dataset. We are providing a description and a discussion of both phytoplankton distribution and for the spectral signal in the section 4.3. of the revised manuscript.

---

## Referee Report (RR1)

Linking satellites to genes with machine learning to estimate phytoplankton community structure from space
El Hourany et al.

Review of revised manuscript

The work of El Hourany and co-authors presents a machine learning approach (specifically, Self-Organizing Maps) to estimate the relative Chla contribution and cell abundances of seven major taxonomic phytoplankton groups. The results of the trained model are applied to global satellite data, and in turn compared to both a previous SOM model developed using pigment rather than omics-based biomarker data, and a separate DPA-based approach. The study is novel in its use of phytoplankton gene information to train an ML model for assessing phytoplankton community structure from space, and the authors have clearly put thought into comparison with other approaches, and to how uncertainties also play a role in the results. After reviewing the manuscript (and in the context of previous reviewer comments and subsequent revisions), I believe the manuscript is publishable following some minor revisions and corrections. Thanks to the authors for their work on this topic.

General comments

I appreciate the background and description of functional types, the DPA and pigment-based groups in the Introduction text. However, the phrase "phytoplankton functional types" is used several times in the document, when in fact what is meant is phytoplankton taxonomic groups. Although "functional types" has been used rather loosely in the literature, the term "functional" indicates biogeochemical function (e.g. calcifiers, silicifiers), whereas phytoplankton of different sizes or even different taxonomic groups may serve the same ecosystem function. Therefore, I strongly encourage the authors to instead use the phrase 'phytoplankton taxonomic groups' when that is actually what is meant, or when referring more broadly to the variety of phytoplankton, 'phytoplankton community composition/structure'. This attention to phrasing will benefit the community of researches working on the topic of phytoplankton community composition from space in the context of interactions with any potential stakeholders and end-users.

Could you comment on the fact that the *psbO* is a proxy of individual cells, but the chain-forming phytoplankton types (e.g., *Chaetoceros*) will contain several, or many, individual cells, and will likely be found in the larger size fractions? For example, if the abundance of diatoms is high in the 20-180 fraction, but the *psbO* represents individual cells (vs. chains), the diatom contribution to Chla could be dramatically overestimated.

In my opinion the text of section 3.4 needs to be revised for clarity. Is it not fully clear what the inputs and targets of the random forest regression algorithm are. The following sentence is not clear: "In the internal node, the selected feature (i.e., pigment in this case) was used to make a decision on how to divide the dataset into separate sets with similar responses in terms of a

given phytoplankton group." Suggest revision to help the reader understand the application of the random forest regression method.

Line 351: as I'm sure the authors are aware, the CHEMTAX approach was developed decades ago to address just this. Although it does have its own caveats, it can be a useful tool to compare against, and recent work to improve it and make it more broadly applicable is worth looking into (see Hayward, Pinkerton, and Gutierrez-Rodriguez. 2023. "Phytoclass: A Pigment-Based Chemotaxonomic Method to Determine the Biomass of Phytoplankton Classes." *Limnology and Oceanography: Methods* 21 (4): 220–41. https://doi.org/10.1002/lom3.10541.) Perhaps this additional analysis is not warranted for this study, but it is worth keeping in mind for future work related to comparison of different approaches used to estimate phytoplankton community structure.

Specific comments

While I'm not aware of the specific journal requirements, numbering the equations would make it easier for the reader to reference the equations within the text for future analyses, etc.

L79: start the sentence with "The" to avoid leading with the gene name.

Line 100: should be GlobColour (with a capital "C"), throughout.

Fig. 1 caption: please define '$D_{RCA}$' and '$D_{ChlF}$' for the reader here

Fig. 2 – it would be valuable to also know the absolute *psbO* values as well – for example, it is true that the Prokaryotes are over-represented in the largest size fraction, but are the absolute quantities of *psbO* very low in that size fraction? I guess more generally – what is the range in absolute quantities of the *psbO* gene across the size fractions?

L116: please define 'CCI'

L130: sentence is awkward as written and ending in "them"; suggest revising to something like "we used two previously published algorithms:"

L149: First sentence does not add anything for the reader.

L 169: is 'variables' here referring to the phytoplankton groups, the satellite-derived parameters, or both?

L 174: what is meant by 'the SOM algorithm that can deal with missing values'? can you give a sentence or two to describe mathematically what is done to account for missing values?

L 184-6: could you add reference(s) here to back up this widespread use of SOM to complete missing data?

L 195: increased from 10 to 1000 neurons at what interval?

Line 313: Curious how you decided on the threshold of 40%?

L 357: typo 'Glocolour' (missing 'b')

L 358: typo 'Fig. 101'

L 371: typo 'Fig. 112'

Figure 9. – suggest including a colorbar to show the number of points per pixel based on the color of the dots on the graph

Figure 12. caption – not clear what is meant by 'original Rrs spectra'

Figure 13. Capitalize the first word of the caption. Could the x-axis of the latitude line graph be revised to label more than just the 10^0 ?

L 448: suggest revision to 'launch of NASA's Plankton, Aerosol, Cloud, ocean Ecosystem (PACE) mission'

L 451: suggest revision to 'the perspective of the PACE mission,'

Alison Chase

---

## Referee Report (RR2)

Review of revised manuscript:
Linking satellites to genes with machine learning to estimate phytoplankton community structure from space
By El Hourany et al.

Thank you to the authors for thorough revisions and thoughtful replies to reviewer suggestions. The following only minor/technical changes; following these changes I believe the manuscript is ready for publication.

Line 43: correct to "in terms of"

L45: I think the parentheses need to be corrected, as-is it is a mix of only around the year and around the full citation.

L88: suggest revision to "a micro-size diatom compared to a pico-sized *Synechococcus*" to remove subjectiveness of "huge" and "tiny"

L90: suggest changing "way greater" to " significantly greater". Also *Syn* needs to be italicized in this line

Figure 1 caption: the second-to-last instance of psbO needs to be italicized

Fig 2 caption: the last instance of psbO needs to be italicized – double check throughout for consistency

L138: suggest revision to "We averaged the output of the four…"

L163: add a space before "Soppa"

L180: is there a specific number you deem a "low number" of pixels?

L 183: no need to re-define Chla here

L 236: I think "real" should be "non-transformed" Chla values?

L348: suggest revision to "observed for diatom Chla fraction,.."

L 359 'Threshold' should not be capitalized. Or, perhaps revise to make the sentence currently within parentheses its own sentence.

L481: "groups; The" – 'The' should not be capitalized

---

## Author Response (AR2)

Dear Editors and Referees,

We would like to express our sincere appreciation for the insightful comments and suggestions that have significantly contributed to improving our manuscript. We have carefully considered the referees' feedback, and in this revised version, we have addressed the concerns and provided comprehensive clarifications as suggested.

Specifically, we have restructured the referencing in line with the referee's recommendations, aiming to provide a more accessible and illustrative discussion rather than relying solely on specific algorithms. Responding to the referee's insightful suggestion, we have introduced metrics to evaluate our SOM methodology, which has shed light on the relative errors inherent in both psbO-based algorithms. This key insight underscores the complexities associated with the errors of SOMRCA in estimating phytoplankton relative abundances when compared to the estimation of Chla fractions per phytoplankton group using SOMChlF. Last, we have discussed the uncertainty associated with SOMRCA and SOMChlF, emphasizing its implications in contrast to previous studies that utilized the DPA approach.

We have taken great care to ensure that the manuscript and the responses provided in this document are aligned and effectively address the concerns raised by the referees.

**Response to referee #1**

Hourany et al. have been developing a machine learning based algorithm trained on several remotely sensed products (RRS, bbp, Kd490, SST, CHL) combined with omics-based biomarker developed from the RV Tara Ocean data set to obtain cell abundance and fraction to total Chla of seven major marine phytoplankton groups. They have evaluated their algorithm with cross-comparison, independent validation and intercomparison to similar satellite products. While I think overall the method development seems to be robust and documented, the manuscript lacks especially:

a)  correctly referencing other work done in the field of phytoplankton measurements, analysis and especially PFT algorithm development,

b)  several details in the two chapters "Materials" and "Methods", and

c)  discussion on their algorithm performance regarding pixel uncertainty, cross-validation, independent validation and intercomparison results.

Below I detail further these shortcomings.

Because of this I think the manuscripts require in these aspects substantial revision before it can become accepted, while most of the other parts can mostly remain.

Detailed comments:

1. It would be good also to have a list of abbreviations in the supplement. There are so many abbreviations used and parameters listed in the manuscript, it becomes confusing.

We added a list of acronyms in the end of the main document (Table 4)

2. Introduction: at several sentences the references provided are not clear or correct or do not merit former work executed in the field:

a) Line 34-35: that is a very sloppy statement "… a range of ecological and biogeochemical problems" What is meant by problems?

We meant by the use of the word "problems" to address various scientific questions

This has been changed in the text to make it clearer:

*This interest has facilitated the integration of the concept of phytoplankton functional types (PFT) and taxonomic groups (PG) into studies exploring various ecological and biogeochemical aspects (Le Quéré et al., 2005; Hood et al., 2006).*

b) Line 39 ff. it is not clear if the methods developed to detect "… abundance of PFT and SC are also meant to be based optical characteristics – since this is clearly stated for the "specific taxa" this should also be clarified here and the references provided then should match the specific method principle. I recommend then to cite here overview papers (see IOCCG 2014, Mouw et al. 2017, Bracher et al. 2017) or at least to put "e.g." since the citations provided are far from complete. In addition, Alvain et al. 2005 and Ben Mustapha et al. 2013 retrieve dominant groups and no abundances, and Chase et al. 2020 method does retrieve PSC from satellite ocean color data, it assessed the diagnostic pigment method based on in-situ data for phytoplankton size classes.

The paragraph has been modified according to the referee's suggestions.

c) Line 48 ff. : should also merit Brewin et al. 2010. A three-component model of phytoplankton size class for the Atlantic Ocean. Ecological Modelling, 221(11), pp.1472-1483. – I would put "e.g." since this list is far from complete!

The statement has been modified accordingly.

d) Line 52 should reference to Brewin at al. 2015 not 2014!

The reference has been modified accordingly.

e) Line 62 (also Methods chapter 2.3.1): You say you downloaded the Xi et al. product from the Copernicus website – if it was after July 2021, it most probably is the product based on Xi et al. 2021 which includes the SST as variable to constrain the algorithm.

We apologize for the confusion, we indeed used the newest version of Xi et al (Xi et al., 2021).

Therefore, we rectified the description of this product.

3. Material & Method sections:

a) a flow chart (Fig. 4) is provided for the SOM DRCA & DChlF data sets – however, everything else connected to methods applied in study is lacking. Since you did many different other parts (DPA three coefficients averaging for HPLC data global and Tara, uncertainty assessment, satellite product intercomparison, cross validation, etc.) – it would be good to have an overview.

To enhance the transparency and comprehensibility of the methodology, we have updated the flowcharts according to the referee's suggestion, providing a detailed overview of each step in the algorithm.

To manage the complexity of the process, we have introduced three sub-flowcharts: one outlining the general training procedure, another focusing on the parametrization of the Self-Organizing Map (SOM) and the selection of variables within the training procedure, and a third delineating the operational phase. These figures have been included in the supplementary materials, and in the main text, a statement has been added referring to these flowcharts.

These flowcharts are intended to provide an accessible and comprehensive understanding of the algorithm, ensuring that readers can navigate and comprehend the various stages of our approach.

b) Chapter 2.1.1- line 75 ff.: It is not clear why stations are discarded when not all 5 size fractions were contained in a station sample – for me it does not make sense from an ecological standpoint. In addition, you do not mention how many stations were then excluded.

We apologize for this error, we indeed verified and there were no stations among the 145 stations that have been discarded. All stations have been utilized and size fractions were aggregated into an average value of relative psbO read per group. We have described the source data in a supplementary figure S1.

*Added: Among the 210 Tara Oceans stations, 145 stations sampled psbO reads in different ocean regimes from oligotrophic to eutrophic waters (Chla from 0.01 to 10 mg.m-3, median at 0.3 mg.m-3, from 2009 to 2013. Seawater samples were filtered in order to differentiate five planktonic size fractions (0.22-3um, 0.8-5um, 5-20um, 20-180 um, 180-2000 um). For the purpose of this study, we pooled the five size fractions into a single aggregated sample.*

Also add the information what exact values for the weights were taken for each size fraction to obtain their chl-a fraction.

First, as mentioned in the text, all phytoplankton groups in each size class were weighted equally by the mid-value of the size range, i.e., x0.9 for the first size class [0.6-1.2], x2.9 for the [0.8-5] size class, x12.5 for the [5-20] size class, and last x100 for the [20-180] size class. Applying equation 1 pools all size fractions per group while considering the psbO read values and the size factors mentioned above.

Why do these weight values make sense for the conversion?

The psbO measurements are proxies of relative cell abundance since this protein-encoding gene is generally present as a single-copy and is found in all phytoplankton groups. For example, if we take a huge diatom compared to a tiny Synechococcus, both have 1 psbO gene and therefore are counted as 1 within the psbO quantification. However, we know that a diatom's Chla content is way greater than that of Synechococcus (Agustí, 1991; Fujiki and Taguchi, 2002; Dairiki et al., 2020; Bock et al., 2022). This is where the conversion via size-dependent weights is essential in the case of Chla content estimation.

In Line 82 it is not clear what 5% here means – relative to the total abundance in each size class or for each size class?

5% of the total cell relative abundance among all size classes.

**In response to the questions of the referee, we decided that it is essential to add further clarification on this aspect to the manuscript (see section 2.1.1)**

c)  Chapter 2.1.2, line 205: Add more information by providing exactly the 11 bands used from 412 to 670 nm from the RRS data set.

The 11 Rrs bands were: 412, 443, 469, 490, 510, 531, 547, 555, 620, 645, and 670 nm. Added accordingly in the text.

d)  Chapter 2.2: Overall, I wonder why not much more HPLC data have been used for your algorithm validation. E.g., you cite Xi et al. 2020 – then you should be aware of the much bigger pigment data set used in this work (taking advantage of the compilation in Losa et al. 2017).  Further check also identification on the error in LTER Palmer HPLC data in Xi et al. (2021) – it may also affect already your compiled data set.

Thank you for pointing out the existence of a more extensive dataset in Losa et al., 2017. It is important to clarify that the HPLC dataset was not solely employed for validation purposes, but rather for comparing the estimations of phytoplankton groups using two different methodologies: the DPA and the *psbO*-derived satellite algorithm. These methods are based on distinct assumptions and resolutions of phytoplankton groups. Using the DPA as a direct validation for the *psbO* data presents challenges. The estimation of phytoplankton groups using pigments is inherently imperfect and relies on assumptions that introduce considerable variability and bias in determining the contribution of specific pigments to the assessment of phytoplankton groups.

During the first phase of the review process, referee #2 raised concerns regarding labeling this comparison as an independent validation and suggested a thorough review of the validation scheme of our algorithm. Consequently, in the revised version of the paper we have re-evaluated the validation process of our algorithm as recommended by referee #2 while introducing a test set validation as explained in the paper.

Therefore, the HPLC dataset was primarily utilized to compare different levels of information and demonstrate the agreement between HPLC and *psbO* data. The database we employed is deemed sufficient to address the questions posed.

NB: Both Losa et al., 2017 and our compiled HPLC database share many common sources, particularly the compiled database of MAREDAT, which constitutes a major part of both datasets.

Finally, before your paper becomes accepted, the compiled HPLC data set with the diagnostic pigments, total chl, and retrieved PFT chl-a conc. should be made available to the readers (e.g., by storage in a public repository).

All *psbO*, HPLC, and satellite matchups datasets will be made available to the community in a public repository, alongside the SOMChlF and SOMRCA algorithms with their operational functions. A statement will be added in the acknowledgment.

e) Chapter 2.2: Why did you choose to apply for the dpa method using the 3 sets of coefficients proposed by Uitz, Brewin, Soppa and that then taking from these calculations the average fraction. You should at least somewhere discuss why you followed this method, instead of just using the coefficient proposed by one of author (I would rather recommend then the newest citation – actually newer ones have been published since then).

The selection of the three sets of coefficients was based on their estimation using global HPLC datasets. While there are newer data sets available, such as those derived by Brewin et al. (2017) and Chase et al. (2020), it is important to note that these are primarily developed using HPLC data at regional or basin scales, as demonstrated in the case of the northern Atlantic Ocean in the examples mentioned.

We are grateful to the referee for bringing the study of Losa et al., 2017 to our attention. In this revised version, we have utilized the coefficients tuned on a global HPLC dataset by Losa et al. (2017).

A thorough examination of the values assigned to the coefficients by these four studies reveals disparities that do not consistently align across all pigments. Notably, while the coefficients for diatoms exhibit similarity across the four sets, differences arise, for instance, in the case of prokaryotes, only Brewin et al. (2015) and Uitz et al. (2006) show close coefficients associated with Zea, while in the case of haptophytes, where only Brewin et al. (2015) and Soppa et al. (2014) estimates similar coefficients attributed to 19HF. The discrepancies can be attributed to variations in the datasets utilized for coefficient estimation and differences in the methodologies employed.

To ensure the robustness of the results and to account for the diverse outputs stemming from the utilization of these coefficients, we opted to compute the average of the outputs from the three sets of coefficients in the previous version, now from four sets of coefficients while adding Losa et al., 2017.

**Added:** *An examination of the values assigned to the coefficients by these four studies reveals disparities that do not consistently align across all pigments. Notably, while the coefficients for diatoms exhibit similarity across the four sets, differences arise, for instance, in the case of dinoflagellates, only Brewin et al. (2015) and Uitz et al. (2006) show close coefficients associated to Perid, while in the case of haptophytes, where Brewin et al. (2015), Soppa et al. (2014) and Losa et al., (2017) estimates close coefficients attributed to 19HF. The discrepancies can be attributed to variations in the*

*datasets utilized for coefficient estimation and differences in the methodologies employed. We chose to do an average of the output of the four sets of coefficients to increase the robustness of the results while considering the different outputs of the utilization of these coefficients.*

f)    Chapter 2.3.1: mind to check if the basis of the CMEMS global PFT product is really Xi et al. 2020 (see comment 2e)– add also the version number of the product in the description. In any case the product is not provided from 1997, but only from 2002 onward. In any case you description that this algorithm uses 15 bands is not correct at all. Please carefully check and provide a correct description.

We apologize for the confusion, we indeed used the newest version of Xi et al.

Therefore, we rectified the description of this product.

**Added:** *This Globcolour product contains the concentration of each phytoplankton functional type (expressed in terms of Chla concentration fraction) based on the Xi et al., 2021 algorithm, processed from 2002 to the present. This algorithm estimates the Chla concentration of diatoms, dinoflagellates, haptophytes, green algae, and prokaryotes. The algorithm was implemented using HPLC-based phytoplankton groups using the DPA approach (Losa et al., 2017, Soppa et al., 2014) merged to OC Rrs products (412, 443, 490, 510, 531, 547, 555, 670, and 678 nm) and accounting for the influence of SST on the derived PFT quantities (product number: OCEANCOLOUR\_GLO\_BGC\_L3\_MY\_009\_103).*

g)    Chapter 2.3.2: it is unclear if also the PFT-chla derived from SOM predicted pigments using Hourany et al. 2019a have been produced by using the average value from applying in the DPA the 3 sets of coefficients proposed by Uitz, Brewin, Soppa. Please clarify.

Indeed, the SOM-Pigment outputs from El Hourany et al., 2019 were derived using these 3 sets of coefficients proposed by Uitz et al., 2006, Soppa et al., 2014, and Brewin et al., 2015.

We have added this in the manuscript in section 2.3.2.

h)    Chapter 3.1 – line 162: it seems except for matching the data based on 3x3 pixel box +/-1 day no further criteria to select "valid" matchups has been used. Protocols recommend that at least 50% of the pixels are valid (unflagged) and the coefficient of variation is within 20% (e.g., see EUMETSAT protocol: https://www.eumetsat.int/media/44087 ). Can you provide more details or comment why no further quality control had been applied.

Indeed, to extract the match-up for a given observation, a 3x3 pixel box was employed, centered around the observation's coordinates on the same day. The average of the non-outlier pixels was computed. If this approach was unproductive due to a low number of pixels within the 3x3 box or the absence of any pixel, a 3x3 pixel extraction was performed for the adjacent days (+1 and -1). However, we did not enforce any additional strict protocols as per the EUMETSAT protocol, as only a small number of valid matchups were anticipated.

To our knowledge, the *psbO* gene database is a valuable source that provides complete information about the relative phytoplankton cell abundance across 7 taxonomic groups. Thus, the intrinsic value of this database is significant. While recognizing the importance of the EUMETSAT protocol in ensuring data quality and homogeneity in match-up exercises, it is important to highlight that our methodology, based on Self-Organizing Maps (SOM), has proven to be effective in reducing noise through vector quantization. Added to that, given the operational nature of the method and the coherence of results from cross-validation and tests, we believe that the evidence showing that this protocol is convincing.

It is imperative to emphasize that any future generation of *psbO* datasets should adhere to the EUMETSAT protocol or other masking protocols adopted by the OC community in the future.

Following these match-up exercises, we performed a baseline comparison between in-situ Chlorophyll-a (Chla) and satellite-derived Chla. This comparison is deemed satisfactory, with an error rate of 33%.

*Added: To extract the match-up for a given observation, a 3x3 pixel box was employed, centered around the observation's coordinates on the same day. The average of the non-outlier pixels was computed. If this approach was unproductive due to a low number of pixels within the 3x3 box or the absence of any pixel, a 3x3 pixel extraction was performed for the adjacent days (+1 and -1). Following these match-up exercises, we performed a baseline comparison between in-situ Chlorophyll-a (Chla) and satellite-derived Chla. This comparison is deemed satisfactory, with an error rate of 33%.*

i)    Chapter 3.2.2 – line 227 ff: Since you noticed that using 670nm in the algorithm did not improve it, why did you keep it? Further, in Line 230 the reference of Xi et al. (2015) is not suited since the paper is focusing on simulated data sets across many (all) water types – probably much better to cite here Torecilla et al. (2011) or Taylor et al. (2011)

where the HCA method (or Alvain et al. 2005 with Physat) has been applied to RRS data from the open ocean in order to derive information on phytoplankton community structure.

As previously mentioned in the manuscript, the 670 nm band was excluded from the algorithm. However, during the initial round of the review process, one of the referees emphasized the importance of utilizing the remote sensing reflectance spectrum, extending up to the near-infrared range. In response to this suggestion, we referenced the work of Xi et al. (2015), as recommended by this referee.

We simplified the explanation regarding the final selected bands in our algorithm to address potential queries that readers might have on this matter while omitting the discussion about the RRS at 670 nm that was not included in the algorithm.

*Added: The choice of Rrs bands aligns with previous work conducted on the PHYSAT method by Alvain et al. (2005) and Ben Mustapha et al. (2013). The PHYSAT method utilizes reflectance anomalies in the same four selected bands to identify dominant phytoplankton functional types. In the clear open ocean, the information contained in the remote sensing reflectance (Rrs) bands beyond 555 nm is limited due to the strong absorption by water (Torrecilla et al., 2011; Taylor et al., 2011). It should be noted that the Rrs bands selected are commonly measured by all sensors used to build the Rrs product of Globcolour. This overlapping of different sensors enhances data availability and coverage, thus increasing the importance of these Rrs bands within the initial dataset.*

j)    Chapter 3.2.4: I missed a discussion about the input data uncertainty influencing the uncertainty of the retrieved PFT products (should be put in chapter 4).

Currently, no comprehensive uncertainties encompass all the associated steps in the quantification of *psbO*, including filtration, extraction, and the accuracy of *psbO*-based analyses. To address these uncertainties, a statement has been included in section 3.2.4 to provide further elaboration on the complexities and potential variations in the quantification process of *psbO*.

*Added: However, we should acknowledge the importance of addressing the uncertainties in the psbO measurements and their potential impacts on the algorithm's outputs, that are not taken into account in this study. This exclusion is primarily due to the absence of a comprehensive framework that accounts for all the associated steps in the quantification of psbO, including aspects such as filtration, extraction, and the accuracy of psbO analysis. Pierella Karlusich et al. (2022) conducted a thorough comparative study, evaluating psbO quantities against data obtained from confocal and optical microscopy, as well as cytometry, revealing an agreement of 70% (Spearman's Rho =0.64–0.71, p-value <.001).*

*However, it is essential to recognize that, like psbO, every quantification method is subject to uncertainties stemming from the various steps of the quantification process, emphasizing the necessity of comprehensive assessments within every in-situ measurement protocol.*

k) Chapter 3.4: The cross-validation results should also provide information of the mean or median relative deviation (MRD) in order to be comparable to other approaches (e.g., Xi et al. 2020, 2021, Lange et al. 2020) – it would be good to have here more statistical measured.

MRD has been incorporated across the study.

All the metrics, old and newly added, were further discussed in text section 4.1. Section 4.1 was modified according to the newly added information.

Direct comparisons with other approaches based on error values remain challenging. It is essential to recognize that this algorithm is rooted in a genomic dataset, delineating taxonomic groups differently from the HPLC DPA method, as exemplified in studies such as Xi et al. (2020, 2021) and Lange et al. (2020). A notable bias between HPLC DPA-derived PFT and *psbO*-derived PFT groups arises from the contrasting definitions of these PFT groups. As well as the differences in PFT group definition, the quantified errors also show the sensitivity specific to each algorithm and methodology followed and can be associated to the coherence of the dataset used in the study.

The comparability of these methods lies within the patterns observed at a global scale and the seasonal variations, enabling to highlight the convergence and divergence between the DPA and *psbO* methods.

***Added:*** *The cross-validation and test exercises demonstrated an average R2 of 0.68 for SOMRCA and 0.74 for SOMChlF across all phytoplankton groups (Fig. 5, table 3). Aggregating all Chla fractions showcased a satisfactory agreement between estimated total Chla and in-situ values (R2= 0.83), indicating the preservation of the initial phytoplankton quantity expressed in total Chla. For SOMRCA, the RMSE ranged between 2% and 23% in the test set and between 2% and 19% in cross-validation. The highest errors were observed for Prokaryotes, reaching 24% due to their high relative cell abundance in the initial dataset. In the case of SOMChlF, the RMSE ranged between 0.02 and 0.24 mg m-3 in cross-validation and 0.02 and 0.31 in the test set, with the highest error associated with the estimation of Chla, stemming from the cumulative Chla fractions of phytoplankton groups. Notably, the largest RMSE among phytoplankton groups was observed for the Diatom Chla fraction, attributed to their substantial Chla content and its exponential relationship with total Chla. The MRD highlighted a distinct contrast between SOMRCA and SOMChlF performance. Notably, SOMRCA exhibited a significantly higher*

*median relative deviation, approximately three times that of SOMChlF's MRD. The MRD for SOMRCA fluctuated between 0.36 and 0.81 for cross-validation and between 0.28 and 0.92 for the test set, with Dinoflagellates exhibiting the highest MRD. In contrast, SOMChlF's MRD per group ranged between 0.13 and 0.24 for phytoplankton Chla fraction and 0.33 for Chla in the test set. This discrepancy emphasizes the complexity of determining the phytoplankton community structure in terms of relative cell abundance, indicating the likelihood of diverse community structures responding to the same satellite-derived environmental context.*

*Added: Uncertainty values reached 30% relative cell abundance for SOMRCA and 0.15 mg m-3 of Chla for SOMChlF, revealing distinct regional patterns in both cases. Notably, the observed uncertainties generally aligned with the concentration gradient in Chla fraction and cell abundance per group. The uncertainty associated with SOMRCA's outputs corresponded to the high relative deviation noted in the test and cross-validation, suggesting the potential acceptance of multiple community structures represented by the neurons of SOMRCA for a single satellite pixel, thus contributing to increased uncertainty levels. Regions at high latitudes exhibited the highest uncertainties for diatoms, green algae, and haptophyte relative cell abundances, while the Southern Ocean displayed heightened uncertainties specifically for prokaryotic cell abundance. The increased uncertainty within the Southern Ocean, particularly for prokaryotes, could be attributed to the limited sampling conducted in this geographical region. This limitation resulted in a notable dissimilarity between satellite data collected in this area and the data sampled in the initial dataset, aligning with the findings of the reliability index. This finding is consistent with the documented very low abundance of cyanobacteria in the Southern Ocean (Flombaum et al., 2013), which may contribute to heightened model uncertainty for this particular region.*

4. Section Results and Discussion

a)  Figure 7 caption: provide n (number of observations) for both data sets, the cross-val set and the test set. As stated above also show (and discuss) results for RMSD and MRD since $R^2$ is not a very robust measure of accuracy of a product. For the PG-Chla comparisons it should be clearly stated in chapter 3 that $R^2$ results from calculations based on log-transformed data, while MRD and RMSD are based on non-log-transformed data.

MRD values are provided in Table 3. In the related Figure 7 (Figure 5 in this new version) we added in the caption to refer to Table 3 for further metrics. We added the n values in the caption of Figure 5 and Table 3.

It is clearly stated in section 3.2.1, Figure 5 and Table 3 that $R^2$ and MRD result from calculations based on log-transformed data, and RMSE is based on non-log-transformed data.

We did not put the RMSE nor the MRD results in Figure 5 due to the overcrowding of the image.

b)    Line 320ff: I think it is difficult to understand what is presented in Figure 8 and discussed here and no values specific for each group and separately for chla-fraction and abundance                                                        are                                                        provided.
The results in fFgure 8 (now Figure 6 in this version) present a pixel-by-pixel indicator of the applicability of the method. As described in section 3.2.4, this indicator is acquired upon comparing the values for each parameter in a pixel to the initial data set used to train both SOM algorithms. It shows the flaws that are brought by the low coverage of the initial data set in certain regions of the global ocean. It is not in any way an uncertainty estimate, but a potential confidence/validity mask that can be associated to the outputs. This has been explained and discussed in the text in section 4.1.

Your pixel-by-pixel uncertainty assessment in terms of values and what it actually considered should be compared to other PFT/PSC algorithms results (e.g. see Brewin et al. 2017, Xi et al. 2021, Lange et al. 2021) - probably in chapter 4.3.

A comparison of uncertainties has been added in section 4.4. However, one may note that, as mentioned in the previous answers, this algorithm is based on a genomic dataset, with a different definition of the taxonomic groups than seen in the HPLC DPA method and using different algorithms.

***Added:*** *Upon comparing the uncertainty patterns with those observed in Xi et al. (2021), similar trends were identified for the Chla fraction of eukaryotic phytoplankton, displaying consistency in following the Chla concentration gradient as seen in our study. Notably, regions such as the gyres exhibited lower uncertainties, whereas higher uncertainties were evident in high-latitude regions and marginal seas. Conversely, when examining the uncertainty in the retrieval of prokaryote Chla by Xi et al. (2021), lower uncertainties were noted in polar regions, contrasting with higher uncertainties observed in low-latitude regions. Similarly, in Brewin et al. (2017), the uncertainty maps for diatoms and dinoflagellates depicted distribution patterns akin to our uncertainty estimates in the North Atlantic Ocean.*

*The noted coherence in uncertainty patterns between HPLC-based products and our psbO-based product can be attributed to the direct relationship between DPA pigment concentration and total Chla, as well as between psbO-derived Chla fractions and total Chla. Consequently, similar patterns in predictions, as well as in the uncertainties, emerge.*

*However, addressing the similarities and differences between the outputs of the above-cited methods referring to the same phytoplankton group is not a straightforward task. These methods are based on distinct assumptions and resolutions of phytoplankton groups; The estimation of phytoplankton groups using pigments is inherently imperfect and relies on assumptions that introduce considerable variability and bias in determining the contribution of specific pigments to the assessment of phytoplankton groups. For instance, several studies showed that the DPA approach tends to overestimate diatoms (Brewin et al., 2014, Chase et al., 2020). This approach may compromise the relevance of satellite images when used. However, the added value of such an approach resides in the availability of the large HPLC dataset, which allows the development of robust algorithms. On the other hand, the method described in this paper and the generated outputs are based for the first time on a complete and harmonized database of phytoplankton taxonomic community structure on a global scale; an approach that provides an unbiased picture of phytoplankton cell abundances. At this time the major limitation of this approach is the low number of observations from which the metric has been derived.*

c)   In addition, in chapter 4.1 and 4.2 a discussion of your two gene-SOM algorithms performance in respect to cross-validation (e.g. as done in Brewin et al. 2015, Xi et al. 2020, 2021) and independent validation to other PFT /PSC algorithms presented in literature (see Mouw et al. 2017 and search newer literature on PSC algorithms) should be added.

All performance metrics, old and newly added, have been further discussed in text section 4.1. Section 4.1 was modified according to the newly added information.

d)   Figure 9, also add the number of matchups (at least in the figure caption), add also the MRD!

The number of matchups has been added (N=2671) in the caption of Figure 9 and in the text (Figure 7 in this version) and the MRD values have been added to the figure.

e)   Fig. 11 color scale for Chl-a should contain more colors, as in Fig.11 abundance presentation and in Fig. 13, so differences in Chl-a are more visible.

Fixed accordingly.

f) Typos: in line 358 and 370 – this should cite the correct subfigures of Fig. 11.

Corrected

**Response to referee #2**

Review of revised manuscript

The work of El Hourany and co-authors presents a machine learning approach (specifically, Self- Organizing Maps) to estimate the relative Chla contribution and cell abundances of seven major taxonomic phytoplankton groups. The results of the trained model are applied to global satellite data, and in turn compared to both a previous SOM model developed using pigment rather than omics-based biomarker data, and a separate DPA-based approach. The study is novel in its use of phytoplankton gene information to train an ML model for assessing phytoplankton community structure from space, and the authors have clearly put thought into comparison with other approaches, and to how uncertainties also play a role in the results. After reviewing the manuscript (and in the context of previous reviewer comments and subsequent revisions), I believe the manuscript is publishable following some minor revisions and corrections. Thanks to the authors for their work on this topic.

General comments

I appreciate the background and description of functional types, the DPA and pigment-based groups in the Introduction text. However, the phrase "phytoplankton functional types" is used several times in the document, when in fact what is meant is phytoplankton taxonomic groups. Although "functional types" has been used rather loosely in the literature, the term "functional" indicates biogeochemical function (e.g. calcifiers, silicifiers), whereas phytoplankton of different sizes or even different taxonomic groups may serve the same ecosystem function. Therefore, I strongly encourage the authors to instead use the phrase 'phytoplankton taxonomic groups' when that is actually what is meant, or when referring more broadly to the variety of phytoplankton, 'phytoplankton community composition/structure'. This attention to phrasing will benefit the community of research working on the topic of phytoplankton community composition from space in the context of interactions with any potential stakeholders and end-users.

We fully agree with the referee regarding the importance of defining the phytoplankton groups as taxonomic groups rather than functional types.

In response to the referee's suggestion, we revised the manuscript replacing instances of "phytoplankton functional types" with "phytoplankton groups" referring to taxonomic groups.

For this matter, we found it important to clarify this difference in the introduction:

*"Recently, ocean color data have also been used to gain information about phytoplankton communities, such as their size structure, and their taxonomic or functional composition. This interest has facilitated the integration of the concept of phytoplankton functional types (PFT) and taxonomic groups (PG) into studies exploring various ecological and biogeochemical aspects (Le Quéré et al., 2005; Hood et al., 2006). Functional types refer to distinct categories associated with biogeochemical processes (e.g., silicifiers, calcifiers) and physiological adaptations to environmental factors (e.g., light, nutrients, turbulence), or to more practical categories identified through specific analytical techniques (e.g., pigment types) (IOCCG report N 14). On the other hand, phytoplankton groups correspond to taxonomic classes (e.g., diatoms, haptophytes, cyanobacteria). It is important to note that phytoplankton from different taxonomic groups can perform the same ecosystem function, e.g., both diatoms and silicoflagellates can biosilicify but represent different taxonomic groups. Specialized algorithms applied to ocean color data have consequently been developed to detect specific taxa with distinctive optical characteristics, e.g., (Brown (1995); Iglesias-Rodríguez et al. (2002)), or the dominance of phytoplankton functional types (e.g., Alvain et al. (2005)) or the relative abundance of phytoplankton groups and size classes in term of their contribution to the Chla e.g., Hirata et al., (2011), Xi et al. (2020, 2021), and lately, plankton assemblages and communities e.g., Kaneko et al., 2023, (Sathyendranath et al., 2014; Bracher et al., 2017; Mouw et al., 2017)"*

Could you comment on the fact that the *psbO* is a proxy of individual cells, but the chain-forming phytoplankton types (e.g., *Chaetoceros*) will contain several, or many, individual cells, and will likely be found in the larger size fractions? For example, if the abundance of diatoms is high in the 20-180 fraction, but the *psbO* represents individual cells (vs. chains), the diatom contribution to Chla could be dramatically overestimated.

The inspection of microscopy images from the same size-fractionated samples showed that long chains (such as those from *Chaetoceros*) are frequently found as shorter fragmented chains and individual cells that pass through small mesh sizes (e.g., Pierella Karlusich et al 2021 *Nat Comm* 12: 4160).

In my opinion, the text of section 3.4 needs to be revised for clarity. Is it not fully clear what the inputs and targets of the random forest regression algorithm are. The following

sentence is not clear: "In the internal node, the selected feature (i.e., pigment in this case) was used to make a decision on how to divide the dataset into separate sets with similar responses in terms of a given phytoplankton group." Suggest revision to help the reader understand the application of the random forest regression method.

We admit that the statement the referee pointed out and the paragraph dealing with the random forest was lacking clarity. We have added further information to the text and reformulated our ideas as follows:

*Each phytoplankton group's psbO abundance was associated with its corresponding HPLC pigment measurements performed on the same Tara Oceans station. The ability of pigments to predict a specific phytoplankton group was evaluated using a bagged random forest algorithm (number of learners set to 200), following the permutation-based importance method.*

*Using this method, a pigment composition of the seven major phytoplankton pigments cited in Table 1 was tested to predict the abundance of each of the seven psbO-derived phytoplankton groups and estimate their importance relative to each group. The concentration of each pigment was converted in terms of pigment ratios, a ratio relative to the sum of all pigment concentrations, and in parallel, the psbO-derived relative abundance was used.*

*The bagged random forest algorithm is a set of decision trees, each constituted of internal nodes and leaves. Within the internal nodes, the algorithm uses pigment data as the predictor variable to partition the dataset into subsets based on pigment characteristics. These subsets are then utilized to predict the abundance of specific phytoplankton groups, enabling effective analysis of the importance of pigments to describe the variability of a phytoplankton group. Since this algorithm is used in a case of regression, the training is done while minimizing the error between the psbO-derived phytoplankton group abundance and the predicted one. The permutation-based importance method will randomly shuffle each pigment and compute the change in the model's performance to predict the abundance of a phytoplankton group.*

Line 351: as I'm sure the authors are aware, the CHEMTAX approach was developed decades ago to address just this. Although it does have its own caveats, it can be a useful tool to compare against, and recent work to improve it and make it more broadly applicable is worth looking into (see Hayward, Pinkerton, and Gutierrez-Rodriguez. 2023. "Phytoclass: A Pigment- Based Chemotaxonomic Method to Determine the Biomass of Phytoplankton Classes." *Limnology and Oceanography: Methods* 21 (4): 220–41. https://doi.org/10.1002/lom3.10541.) Perhaps this additional analysis is not warranted for this study, but it is worth keeping in mind for future work related to comparison of different approaches used to estimate phytoplankton community structure.

Thank you for your insightful comment regarding the CHEMTAX approach. We have noted your point and acknowledge the significance of evaluating various methodologies in this domain. In future work, it is indeed our intention to make further attempts to define a consensus from different phytoplankton group identification methods, but we consider that this goes beyond the scope of the current manuscript.

 Specific comments

While I'm not aware of the specific journal requirements, numbering the equations would make it easier for the reader to reference the equations within the text for future analyses, etc.

We added numbering for each equation as suggested.

L79: start the sentence with "The" to avoid leading with the gene name.

Added accordingly.

Line 100: should be GlobColour (with a capital "C"), throughout.

Modified accordingly throughout the manuscript.

Fig. 1 caption: please define 'DRCA' and 'DChlF' for the reader here

Added in the caption.

Fig. 2 – it would be valuable to also know the absolute *psbO* values as well – for example, it is true that the Prokaryotes are over-represented in the largest size fraction, but are the absolute quantities of *psbO* very low in that size fraction? I guess more generally – what is the range in absolute quantities of the *psbO* gene across the size fractions?

The *psbO* can be used to estimate **absolute** cell abundances with careful normalization and quantitative DNA extraction methods. In the current study, we did not attempt to do it because the metagenomic sampling from *Tara* Oceans was not specifically designed to quantify metagenomic signals per seawater volume due to the lack of "spike-ins" (e.g., DNA internal standards). It's well established that there is an inverse logarithmic relationship between plankton size and abundance (Belgrano et al., 2002; Pesant et al., 2015), so small size fractions represent the numerically dominant organisms in terms of cell abundance (albeit not necessarily in terms of total biovolume or biomass).

We can still express *psbO* abundance in rpkm (reads per kilobase covered per million of mapped reads) to normalize the *psbO* signal by the sequencing depth. There is a decrease in rpkm values towards the larger size fractions, probably explained by the

increase in genome size and complexity in larger size fractions. In addition, prokaryotes are dominant in the smaller size fractions while the larger fractions are characterized by the higher prevalence of eukaryotic phytoplankton.

[Figure]

This figure was added in the supplementary material figure while citing it in the caption of Figure 2.

L116: please define 'CCI'

Defined as a Climate Change Initiative (CCI)

L130: sentence is awkward as written and ending in "them"; suggest revising to something like "we used two previously published algorithms:"

It was changed as suggested by the referee.

L149: First sentence does not add anything for the reader.

Removed

L 169: is 'variables' here referring to the phytoplankton groups, the satellite-derived

parameters, or both?

We chose to normalize every variable, Phytoplankton, and satellite, to reinitialize the weights before using SOM, and to make their values comparable. This has been clarified at the end of the paragraph.

L 174: what is meant by 'the SOM algorithm that can deal with missing values'? can you give a sentence or two to describe mathematically what is done to account for missing

values? Could you add reference(s) here to back up this widespread use of SOM to complete missing data?

We have formulated and briefly described how SOM can deal with missing values while adding some references. this formulation was added to section 3.2.1

*Added: SOMs have been widely employed to complete missing data, utilizing the truncated distance (Folguera et al., 2015; Charantonis et al., 2015; Saitoh, 2016; Rejeb et al., 2022). The truncated distance is defined as a modification of the standard Euclidean distance between two observations that accounts only for the existing components of the vectors. This modification of the distance measure allows for the comparison of observations with incomplete information by considering only the existing components and effectively handling missing data. The SOM algorithm can then use this truncated distance measure in its learning process to complete missing data and integrate incomplete information, enabling more robust analysis and visualization of the data.*

L 195: increased from 10 to 1000 neurons at what interval?

The interval is 10. We have added this information to the text.

Line 313: Curious how you decided on the threshold of 40%?

We are sorry for this mistake, but due to the change of this figure across the review process, we meant to say 60% instead of 40. The 60% threshold was an arbitrary choice, looking at the shape of the value distributions of this index and their spatial patterns.

In terms of calculation, 60% means that almost 3 out of 8 satellite parameters at a certain pixel are considered as an outlier, and therefore the estimated phytoplankton composition might be biased.

**An explanation was added in section 4.1:** *Threshold arbitrarily chosen while evaluating the frequency histogram of this index's values in Fig.6. A value of 60% roughly translates to the exclusion of 3 out of 8 satellite parameters' values considered outliers at a certain pixel.*

L 357: typo 'Glocolour' (missing 'b')

Corrected

L 358: typo 'Fig. 101'

Corrected

L 371: typo 'Fig. 112'

Corrected

Figure 9. – suggest including a colorbar to show the number of points per pixel based on the color of the dots on the graph

We have added a density color bar.

Figure 12. caption – not clear what is meant by 'original Rrs spectra'

We meant to refer to denormalized Rrs spectra, as in the original values. We have modified the caption as follows:

*Relative cell abundances per phytoplankton group and normalized and denormalized Rrs spectra were also derived.*

Figure 13. Capitalize the first word of the caption. Could the x-axis of the latitude line graph be revised to label more than just the 10^0 ?

The figure was modified according to the referee's suggestion.

L 448: suggest revision to 'launch of NASA's Plankton, Aerosol, Cloud, ocean Ecosystem (PACE) mission'

Revised          according          to          the          referee's          suggestions

L 451: suggest revision to 'the perspective of the PACE mission,'

Revised          according          to          the          referee's          suggestions

---

## Author Response (AR3)

Dear Editor,

We express our gratitude to you and the reviewers for providing valuable suggestions and constructive comments that significantly contributed to the improvement of our study and manuscript.

In response to the final suggestions from the reviewers, we incorporated all the recommendations put forth by reviewer #2. Additionally, as a response to reviewer #1, we have included the sources and references of the compiled HPLC database in the supplementary document.

Moreover, we are currently in the process of publishing the datasets generated during the course of this study, along with the SOM-psbO operational algorithms. These resources will be accessible on Zenodo under the reserved DOI: https://doi.org/10.5281/zenodo.10361485. The data availability section in the paper has also been updated to show this information.

We sincerely appreciate the interest shown in our paper by both you and the reviewers.

Best regards,

On behalf of the co-authors,

Roy El Hourany